# Identifying associations of de novo noncoding variants with autism through integration of gene expression, sequence, and sex information

Runjia Li[1,2] and Jason Ernst[1,2,3,4,5,6,7*]

*Correspondence:
jason.ernst@ucla.edu

[1] Bioinformatics Interdepartmental Program, University of California, Los Angeles, Los Angeles, CA 90095, USA
[2] Department of Biological Chemistry, University of California, Los Angeles, Los Angeles, CA 90095, USA
[3] Eli and Edythe Broad Center of Regenerative Medicine and Stem Cell Research, University of California, Los Angeles, Los Angeles, CA 90095, USA
[4] Computer Science Department, University of California, Los Angeles, Los Angeles, CA 90095, USA
[5] Jonsson Comprehensive Cancer Center, University of California, Los Angeles, Los Angeles, CA 90095, USA
[6] Molecular Biology Institute, University of California, Los Angeles, Los Angeles, CA 90095, USA
[7] Department of Computational Medicine, University of California, Los Angeles, Los Angeles, CA 90095, USA

## Abstract

**Background:** Whole-genome sequencing (WGS) data has facilitated genome-wide identification of rare noncoding variants. However, elucidating these variants' associations with complex diseases remains challenging. A previous study utilized a deep-learning-based framework and reported a significant brain-related association signal of autism spectrum disorder (ASD) detected from de novo noncoding variants in the Simons Simplex Collection (SSC) WGS cohort.

**Results:** We revisit the reported significant brain-related ASD association signal attributed to deep-learning and show that local GC content can capture similar association signals. We further show that the association signal appears driven by variants from male proband-female sibling pairs that are upstream of assigned genes. We then develop Expression Neighborhood Sequence Association Study (ENSAS), which utilizes gene expression correlations and sequence information, to more systematically identify phenotype-associated variant sets. Applying ENSAS to the same set of de novo variants, we identify gene expression-based neighborhoods showing significant ASD association signal, enriched for synapse-related gene ontology terms. For these top neighborhoods, we also identify chromatin state annotations of variants that are predictive of the proband-sibling local GC content differences.

**Conclusions:** Overall, our work simplifies a previously reported ASD signal and provides new insights into associations of noncoding de novo mutations in ASD. We also present a new analytical framework for understanding disease impact of de novo mutations, applicable to other phenotypes.

## Background

The increasing availability of whole genome sequencing (WGS) data is presenting new opportunities to better understand the associations of rare noncoding variants with complex human phenotypes [1–3]. One such phenotype where WGS has been

applied is to study genetic contributions to autism spectrum disorder (ASD). Prior studies using arrays or exome sequencing led to genetic associations with ASD based on common [4] or rare coding [5–10] variants, respectively, but were not able to extensively study the role of rare noncoding variants. A notable dataset that has been used to begin to understand the role of rare noncoding variants in ASD and more broadly psychiatric and other complex diseases is WGS of the Simons Simplex Collection (SSC) cohort [3, 11–14]. This dataset has enabled calling of de novo noncoding variants from WGS of probands and unaffected siblings for more than a thousand families. De novo variants are of particular interest [15] since they correspond to a subset of variants for which selection has not had a chance to exert its effect and thus can enrich for higher impact variants [16].

However, even when focusing on de novo noncoding variants, there remain substantial analytic challenges in interpreting the association of these variants to ASD. In an initial analysis of a subset of the de novo variants in the SSC cohort, many combinations of variant annotations were tested for burden association through the category-wide association study (CWAS) framework [12]. After multiple-testing correction, this did not lead to any significant noncoding associations in either an initial cohort [12] or later analysis of an expanded cohort [13]. However, an alternative analysis based on a risk score did suggest a significant noncoding association signal and further analysis suggested that this signal was associated with promoter regions [13].

A separate analysis of the SSC cohort defined a DNA-based and an RNA-based "disease impact score" (DIS), which were then used to evaluate the contributions of de novo noncoding variants to ASD [11]. The scores were generated based on first training deep neural networks to predict from DNA sequences chromatin accessibility, histone modifications, transcription factor (TF) binding, and RNA binding. The trained neural networks were then used to compute corresponding feature values for individual variants based on their impact on the neural network predictions. Finally, DNA- and RNA-based features were combined into their respective DISs with a supervised classifier trained using disease-curated variants. In addition, a DIS that combines the DNA- and RNA-based DIS scores was also defined. Various subsets of de novo variants from the SSC cohort were then tested to determine if there was a significant difference in DIS distributions of them between proband and unaffected siblings. This led to reported significant associations including notably among a subset of noncoding variants that are near genes that are differentially expressed in brain tissues.

However, given the complexity of the approach and the challenge of interpreting a score based on integrating many different features derived from deep neural networks, we sought to investigate whether simpler and more interpretable approaches could identify a similar or even stronger association signal. Specifically, here we first show that using local GC content is sufficient to obtain similar results about noncoding variant associations previously attributed to deep learning. We then further show that by considering simple additional information not considered in the prior analyses, namely the sex of the proband and the unaffected sibling and whether the variant was upstream or downstream of their assigned genes, we can better isolate the likely source of the association signal for variants near genes differentially expressed in brain. Specifically, we find that the association signal for variants near differentially expressed brain genes appears

to be driven by variants from families with a male proband and female sibling and are upstream of the gene they are assigned.

Given these insights, we develop Expression Neighborhood Sequence Association Study (ENSAS) to extend the analytical approach to also evaluate higher-order *k*-mers and more comprehensively defined gene-expression variant sets. ENSAS identifies sets of variants associated with a phenotype by integrating systematically defined gene expression neighborhoods and sequence context in the form of *k*-mers. We apply ENSAS to analyze the SSC cohort for ASD associations. When considering variants from male probands and female siblings upstream of their assigned genes, ENSAS refines the set of brain-expressed gene sets most strongly associated with the association signal. Gene ontology enrichment analyses suggest the top associated neighborhoods are related to synapses. Using higher-order *k*-mers either does not improve predictive performance or shows at most a limited improvement over local GC content. Finally, we conduct chromatin state analyses of variants in top neighborhoods which suggest that chromatin state annotations from specific epigenomes can predict a substantial portion of the proband-sibling GC content differences. Together our work provides new insights into de novo noncoding mutations associations with ASD and presents a general analytical framework that could be used for other phenotypes.

## Results

### Local GC content largely explains noncoding ASD associations attributed to deep learning

We first investigated if a simpler and more interpretable procedure can yield similar association signals compared to using DIS score. In particular, across the $n = 127{,}140$ de novo variants we observed a strong correlation between local GC content, defined here as the number of G or C bases in the 201-bp window centered at the variant, and both the RNA DIS and the DNA DIS (Spearman correlation $= 0.57$ and 0.72, respectively) (Fig. 1a, Additional file 1: Figure S1a). This led us to ask whether similar association signals could be obtained with just local GC content.

To test this, we conducted comparisons using the same association tests previously reported for (1) 130 curated genomic variant sets (60 for DNA DIS and 70 for RNA DIS) (Additional file 1: Figure S2) and (2) variant sets defined based on proximity to genes exhibiting tissue-restricted expression activity for expression data in 53 tissues or cell types from the Genotype–Tissue Expression (GTEx) project [17] (Fig. 1b, Additional file 1: Figure S1b, Additional file 2: Table S1). Here for the genomic variant set analysis we used local GC content in place of the RNA or DNA DIS and for the GTEx-based analysis in place of the combined DIS. For the GTEx-based analysis following Zhou et al. [11], we only included variants within 100 kbp of a transcription start site (TSS) or intron variants within 400 bp of an exon boundary, which left 71,554 variants. We also conducted additional tests where we excluded both coding and canonical splice site (CSS) variants from the analysis, both of which have well-established associations with ASD [4, 5, 7–10, 14], which left us with 122,807 variants for the genomic variant set analysis and 67,671 variants for the GTEx-based analysis.

Using local GC content, we obtained an association signal with an overall similar distribution of *p*-values as for the DNA DIS across the 60 curated variant sets ($p = 0.16$ by a two-sided Mann–Whitney $U$ test, the same test as in Zhou et al. [11])

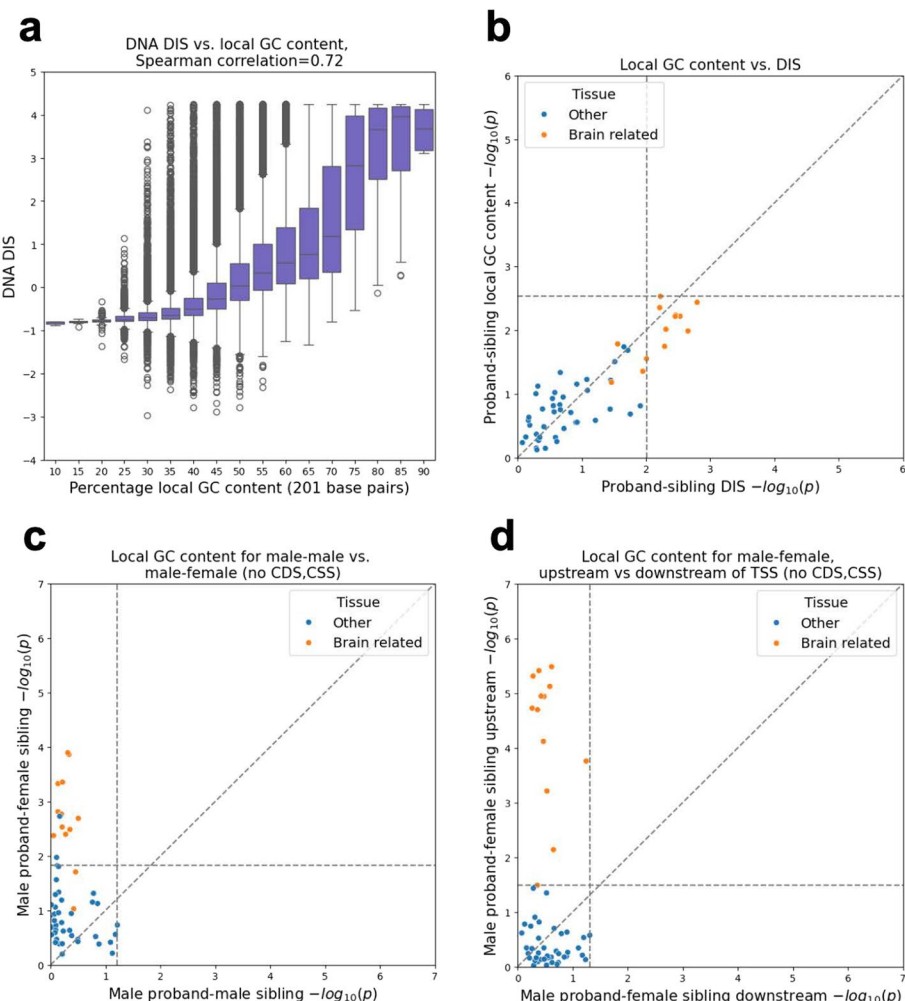

**Fig. 1** DIS vs. local GC content and GTEx analysis. **a** The *y*-axis corresponds to the DNA DIS score and the *x*-axis evenly sized local GC content intervals, each with size 10 (corresponding to 5%, with bin labeled as N representing (N−5)%,N%)). Each interval is represented by a box showing the distribution of its variant's DIS [11]. The boxes correspond to the quartiles, the lengths of whiskers correspond to 1.5 × interquartile range and variants above/below the whiskers were defined as outliers. The reported Spearman correlations were computed between the DNA DIS and local GC content, across all variants. **b–d** Proband-sibling differences for DIS (combined) or local GC content of variants assigned to each of the 53 GTEx tissue or cell types are shown colored based on whether they are brain related or not. Both the *x*- and *y*-axis show -log$_{10}$ *p*-values from one-sided Mann–Whitney *U* tests. Horizontal and vertical dashed lines show *p*-values at FDR threshold of 0.05. Points greater than (but not on) these lines were significant after FDR correction. Diagonal dashed lines show the unit slope. Full list of results available in Additional file 2: Table S1. **b** Proband-sibling differences in local GC content vs. proband-sibling differences for DIS (as conducted by Zhou et al. [11]) **c** Male proband-female sibling pair differences for local GC content vs. male proband-male sibling pair differences for local GC content, with coding and CSS variants removed. **d** Male proband-female sibling pair differences for local GC content in variants < 100 kbp upstream of nearest outermost TSS only vs. in variants < 100 kbp downstream of nearest outermost TSS, with coding and CSS variants removed. Additional related panels can be found in Additional file 1: Figure S1

and the combined DIS across the 53 GTEx tissues ($p = 0.98$). The RNA DIS showed stronger associations compared to local GC content when considering the 70 variant sets based on all variants ($p = 0.009$; Additional file 1: Figure S2a, b), but after excluding coding and CSS variants the difference was no longer significant ($p = 0.16$,

Additional file 1: Figure S2c, d). To more directly test whether the reported DIS significant associations signal could be explained by local GC content, we adjusted the DIS score based on the local GC content (Methods). After adjusting for local GC content and controlling for multiple testing, individual associations were no longer significant (Additional file 1: Figure S1c, Additional file 1: Figure S2e, f).

While these analyses suggest local GC content offers a simpler approach sufficient to produce similar association signals to those reported for DIS, they do not exclude the possibility that there is additional association signal within the DNA sequence beyond local GC content. These analyses also do not explain why local GC content, which can be both biologically significant and a confounder in genomic analyses [18, 19], was sufficient to identify the association signals. In particular, we were interested in understanding this in the context of the result that variants assigned to genes differentially expressed in brain tissue types among a panel of GTEx tissue types had the strongest association signal for ASD.

### Local GC content differences around brain-specific genes are specific to male probands with female siblings

Given the well-established sex bias of ASD cases [20], which was reflected in the SSC cohort with 87% of probands being male compared to 47% of unaffected siblings, we asked if sex differences between probands and siblings might be related to the local GC content differences between their de novo variants assigned to brain-specific expressed genes. Specifically, we repeated the gene expression association tests based on local GC content separately for de novo variants from a family with a male proband and a female sibling ($n = 30{,}430$ from 787 families) and for variants from a family with a male proband and a male sibling ($n = 27{,}251$ from 713 families). We note there was no significant difference between the number of de novo variants per sample between the two groups (two-sided Mann–Whitney $U$ $p$-value $= 0.4$; average 19.33 variants for samples from male proband-female sibling pairs and 19.11 variants for samples from male proband-male sibling pairs). We also note that consistent with the previous analysis all variants that we considered were on autosomes and excluded those variants overlapping repeats [11]. Variants from male proband-female sibling pairs showed noticeably stronger signals than those from male proband-male sibling pairs. For male proband-female sibling pairs, 13 tissues were significant at an FDR threshold of 0.05 with the most significant tissue having a nominal $p$-value of 0.00013. In comparison, no tissues were significant at the same FDR threshold for male proband-male sibling pairs and the most significant tissue had a nominal $p$-value of 0.061 (Fig. 1c). Brain-related tissues were among the top associations for male proband-female sibling pairs with 11 out of 13 tissues significant at an FDR threshold of 0.05 but not for male proband-male sibling pairs where the most significant brain-related tissue had a nominal $p$-value of 0.31 (Fig. 1c). We repeated the analysis for de novo variants from a family with a female proband and a female sibling ($n = 4664$ from 123 families) and for variants from a family with a female proband and a male sibling ($n = 3948$ from 101 families), and in both cases we did not observe any significant signal though we note the much smaller sample size for this analysis (Additional file 1: Figure S1d).

To confirm that the differences between the male proband-female sibling and the male proband-male sibling were not driven by the modest sample size differences, we downsampled the male proband-female sibling variants to 27,251, matching the count of male proband-male sibling variants. We performed this random downsampling and repeated our analyses 10 times. We still observed 11 out of 13 brain-related tissues with significant median $p$-values, with the most significant tissue having a median $p$-value of 0.00031 (Additional file 1: Figure S1e).

To assess the extent to which the observed signal might be associated with general differences between male and female samples, we repeated the analysis by comparing all male siblings to all female siblings, as well as comparing all male probands to all female probands. We did not observe any significant signal after multiple testing corrections, though some associations were nominally significant (Additional file 1: Supplementary text, Additional file 1: Figure S1f). We conducted an additional analysis where we compared male probands restricted to those with female siblings to male probands restricted to those with male siblings, which also did not yield any significant associations after multiple testing corrections (Additional file 1: Supplementary text, Additional file 1: Figure S1g).

Given the male proband-female sibling differences for noncoding variants, we were interested to know if similar male proband-female sibling differences can be found with respect to the number of de novo coding variants or protein-truncating variants (PTVs) (Methods), the latter of which previously showed ASD association signal in the study by An et al. [13] For coding variants, we did not observe any significant proband-sibling difference (Additional file 1: Table S2). For PTVs, we did observe significant proband-sibling differences ($p < 0.05$) when considering all probands and all siblings and all male proband-male sibling pairs, while male proband-female sibling pairs showed an enrichment that did not reach statistical significance (Additional file 1: Table S2). These results suggest the association found for noncoding variants among male proband-female sibling pairs is not directly reflected in the PTV count association signal.

### Male proband-female sibling differences in local GC content are specific to variants upstream of TSSs

In the above analyses, we used the previous assignments of variants to genes based on their nearest representative TSS [11], which did not differentiate variants upstream of a TSS from those downstream. Given the different chromatin environments and mutational processes associated with transcription [15, 21], we reasoned that it could also be informative to analyze the association signal separately for variants upstream and downstream of the TSS of their assigned genes. Specifically, we computed for each variant its position relative to the nearest annotated outermost TSS of a protein-coding gene, restricting to variants within 100 kbp of a TSS, consistent with the distance threshold previously used [11]. We then repeated the above gene expression-based analysis separately for variants from male proband-female sibling pairs that were upstream of their assigned gene and for those downstream. Overall, when restricted to the set of upstream variants ($n = 12,474$), the brain tissues showed even stronger association signals compared to using all variants, with 12 tissues significant at an FDR threshold of 0.05 and the most significant tissue being the anterior cingulate cortex (nominal $p = 3.3 \times 10^{-6}$,

Fig. 1d). Meanwhile, none of the tissues were significant for the downstream variants ($n=16{,}571$), with the most significant tissue being skeletal muscle (nominal $p=0.049$) and the most significant brain-related tissue being hypothalamus (nominal $p=0.057$). Compared to using the combined DIS for this analysis, local GC content showed an overall similar trend but had a more significant $p$-value for 12 of the 13 brain tissues (Additional file 1: Figure S1h). Together, these results suggest the observed difference in local GC content between male proband and female sibling variants near brain-expressed genes can be mainly attributed to variants upstream of TSS.

### Analyses of the brain-related signal with Expression Neighborhood Sequence Association Study

The previous gene expression analyses were limited to using previously defined sets of genes for each of the 53 GTEx tissues, with each set being the genes that in the tissue type had five times its median expression across all tissues. This leaves open the possibility that considering information from additional gene expression data and more comprehensively defined gene expression-based gene sets could capture potential additional signals. Additionally, the previous analyses were limited to only using local GC content in terms of sequence information, leaving open the possibility there is additional association signal information in the sequences that could be captured by higher-order $k$-mers.

We therefore propose Expression Neighborhood Sequence Association Study (ENSAS, Methods), a generalized framework that integrates gene expression neighborhoods and local sequence in the form of $k$-mer counts to discover sets of variants associated with a phenotype (Fig. 2). ENSAS first defines gene-expression-based neighborhoods by considering pairwise gene expression correlations. Specifically, ENSAS uses previously computed gene–gene expression correlations from the Geneshot database [22], which were computed using a large compendium of gene expression datasets collected by the ARCHS4 web resource [23]. Compared to GTEx, ARCHS4 is a more comprehensive and diverse collection of publicly available RNA-seq data, consisting of over 80,000 human samples across different tissues [23]. ENSAS then defines a variant "neighborhood"

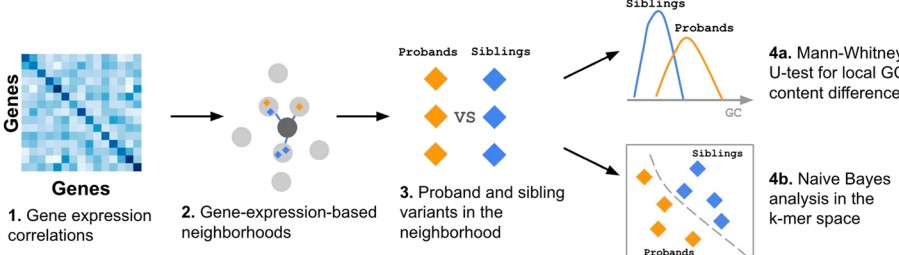

**Fig. 2** Schematic overview of ENSAS. In this schematic overview, the two sets of variants being compared are the proband and unaffected sibling variants. (1) The gene–gene expression correlation matrix is obtained from the Geneshot database [22]. (2) ENSAS identifies the genes with the largest gene expression correlations to a target gene. The neighborhood of the target gene is defined as the top *M* variants assigned to its closest genes based on the correlations, including the variants assigned to the target gene itself if there are any. (3) Each neighborhood consists of two groups of variants which are compared using the following approaches: (4) ENSAS (**a**) performs a Mann–Whitney *U* test between the local GC content of proband and sibling variants, and additionally (**b**) uses a Naive Bayes model to identify sequence differences based on *k*-mers

surrounding each gene, as the variants assigned to the gene itself and the top $M$ variants assigned to a set of closest genes based on the expression correlations (Methods).

After defining gene-expression-based neighborhoods, ENSAS conducts two sets of sequence-based tests for each neighborhood. The first is a one-sided Mann–Whitney $U$ test to test the difference between the local GC content of two groups of variants. The second is a test based on $k$-mer frequencies. Specifically, for this second test ENSAS splits the variants in a neighborhood into equally sized training and testing folds. Using variants from the training fold ENSAS trains a multinomial Naive Bayes classifier with uniform class priors to predict a variant's group label. The features to the classifier are the $k$-mer counts within a $L$-bp sequence window centered on the variant. We used the Naive Bayes classifier as it is suitable for small sample sizes, and similar formulations of $k$-mer-based Naive Bayes classifiers have previously been used for sequence classification tasks [24–26]. ENSAS then uses the trained classifier to score each variant in the testing fold, where each score represents the variant's relative likelihood of being in a target group. ENSAS then performs a one-sided Mann–Whitney $U$ test to test if the target group has higher scores than the non-target group. For comparison with $k$-mer results, ENSAS also conducts the local GC content analysis on the testing fold only.

Overall ENSAS can be seen as an extension of the previous GC content analysis with gene sets defined based on GTEx expression data, where an integration of $k$-mer frequency is tested in addition to GC content and the gene sets are neighborhoods surrounding each gene defined by gene expression correlations.

### ENSAS captures proband-sibling local GC content differences in specific gene-expression-based neighborhoods

We applied ENSAS to the SSC cohort, where the two groups were proband and sibling variants. For this application, we set the sequence length $L$ to 201 bp, the neighborhood sizes $M$ to 1000 variants (mirroring approximately the mean GTEx variant set size that is 1048, Methods) and we used values from 1 to 7 for the length of $k$-mers.

We applied ENSAS restricted to variants from male-proband female sibling pairs that were within 100 kbp upstream of their nearest outermost TSSs (henceforth referred to as "M-F upstream" variants) considered above, leaving us with $n = 12{,}293$ variants. We tested a total of 29,820 expression neighborhoods and after a Bonferroni-based multiple testing correction we observed 28 neighborhoods with a significant difference ($p \leq 1.7 \times 10^{-6}$) in local GC content between proband and siblings. On average, each pair of these 28 neighborhoods showed considerable overlap, with a mean sharing of 705 out of 1000 variants. The top neighborhood consists of variants assigned to the closest genes surrounding OPCML, a synaptic signaling gene [27], with a $p$-value of $1.3 \times 10^{-7}$ (Fig. 3a). We note that the Bonferroni-based correction is conservative in this setting due to the overlap between neighborhoods and that ENSAS offers an alternative permutation-based multiple testing correction method though with a greater computational cost. Using a permutation-based threshold ($p \leq 2.8 \times 10^{-3}$), we observed 1542 significant neighborhoods when applied to the M-F upstream variants (Fig. 3a, Methods). In the interest of analyzing a more limited set and the most significant neighborhoods, we focused our analyses on the 28 neighborhoods that were still significant after applying a Bonferroni correction.

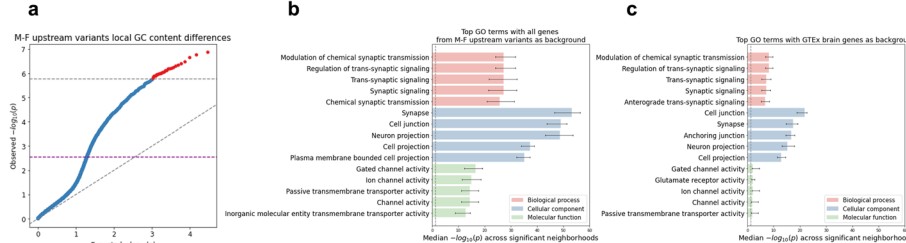

**Fig. 3** ENSAS results for M-F upstream variants in the SSC cohort. **a** QQ-plot of the neighborhood Mann–Whitney *U* p-values for local GC content. Gray horizontal dashed line shows the Bonferroni-based *p*-value multiple testing significance threshold of 0.05/*n* where *n* = 29,820 is the number of neighborhoods. Purple horizontal dashed line shows the permutation-based multiple testing threshold at an FDR of 0.05, with points above these lines significant after correction. Diagonal dashed line shows the unit slope. **b** Top 5 enriched GO terms under each GO category for the 28 significant neighborhoods (shown as red in **a**) using the Bonferroni threshold when using the union of all genes assigned to a M-F upstream variant as background for the enrichment analysis. *x*-axis shows median Benjamini–Hochberg adjusted Fisher's exact *p*-values across significant neighborhoods. Vertical dashed line shows *p*-value significance threshold of 0.05. Error bars show interquartile range across the neighborhoods. **c** Same as **b** but using the union of genes assigned to a M-F upstream variant that are also differentially expressed within any of the 13 brain-related GTEx tissues as background for the enrichment analysis

To biologically characterize the sets of genes associated with the top neighborhoods in the previous M-F upstream variants analysis, we conducted a Gene Ontology (GO) enrichment analysis for genes associated with each of the 28 neighborhoods (Fig. 3a, Methods). For this analysis, we used as background the set of all genes assigned to a M-F upstream variant and separately for each neighborhood tested if the genes in the neighborhood showed enrichment in GO terms compared to the background gene set (Methods). We observed that the most significant term across the neighborhoods was synapse (median Benjamini–Hochberg adjusted $p = 5.8 \times 10^{-54}$, Fig. 3b). We note the *p*-value significance in this analysis is the enrichment of genes in the top neighborhood for the GO category and not the significance of proband-sibling differences for the category. Other terms related to synaptic transmission and transmembrane channels were also highly significant. These classes of genes had previously been implicated based on de novo coding variants in ASD [6, 28].

We next asked whether the GO enrichments are more specific in the top neighborhoods identified by ENSAS or would be expected by also simply considering the set of previously analyzed differentially expressed genes in the 13 brain-related GTEx tissues. To investigate this, we repeated the GO enrichment analysis but now using the set of genes assigned to the M-F variants that are within any of the 13 brain-related GTEx tissues as background. We observed that the top terms were similar as in the previous analysis, with the previously most significant term synapse still highly significant (median Benjamini–Hochberg adjusted $p = 3.5 \times 10^{-18}$, Fig. 3c). The most significant term became cell junction (median Benjamini–Hochberg adjusted $p = 1.8 \times 10^{-22}$, Fig. 3c), which was previously also highly significant ($p = 1.2 \times 10^{-49}$). That the top terms were preserved suggests the possibility that ENSAS identified top neighborhoods may provide more specific biological information compared to the union of differentially expressed genes in the 13 brain-related GTEx tissues.

In addition, we examined the extent to which the top 28 neighborhoods characterized diverse genes and pathways. To do this, we first clustered these neighborhoods using DBSCAN [29] based on their variants (Methods) and then for each cluster repeated the GO enrichment analysis on the union of genes of all neighborhoods in the cluster. We observed similar terms enriched across the clusters, suggesting that the top neighborhoods identified by ENSAS implicate similar biological processes (Additional file 1: Supplementary text, Additional file 1: Figure S3).

### ENSAS Proband-sibling local GC content differences not driven by obvious sequencing batch effects or promoter variants

To exclude the possibility that the observed associations were driven by obvious sequencing batch effects as reflected in different recorded sequencing lanes for probands and unaffected siblings, we separately analyzed two subgroups of samples: samples from pairs that had matching recorded sequencing lane information and those from pairs that did not (Methods). This left us with 968 samples from male proband-female sibling pairs that had matching recorded sequencing lane information and 606 samples from pairs that did not. For each neighborhood, we separately performed proband vs. sibling Mann–Whitney $U$ tests on the two subgroups. When restricted to the samples with matching sequencing lane information, despite the smaller sample size, six significant neighborhoods were still identified as significant, with two of them also significant when considering the full set of samples (Additional file 1). The neighborhoods significant in the full set of samples had a median $p$-value of $1.7 \times 10^{-5}$ in this subset of samples. No significant neighborhoods were identified when restricted to the samples with mismatching sequencing lanes (Additional file 1: Figure S4c) and the overall distribution of the $p$-values appeared relatively uniform (Additional file 1: Figure S4a). However, the neighborhoods significant in the full set of samples had a median $p$-value of 0.0047 in this subset of samples with 7 out of 28 of them among the top 30 most significant neighborhoods (Additional file 1: Figure S4c). Notably, the neighborhood $p$-values from both the lane-matching and lane-mismatching subsets showed high correlation with those from the full set of samples (Spearman's $r = 0.78$ and 0.72 respectively, Additional file 1: Figure S4b,c). Overall, while the signal could be mainly attributed to the subset of lane-matching samples whose comparison would be expected to be less susceptible to technical sequencing confounders, the association from the lane-mismatching samples was in a consistent direction.

We noticed that whether proband and sibling samples had matching or mismatching sequencing lanes was associated with their sequencing phases (Additional file 1: Supplementary text, Additional file 1: Table S3). We thus repeated the tests on each of these phases excluding the pilot phase, which had limited samples. We observed the most significant associations from phase 2, which also had the largest number of lane-matched samples. However, overall associations were enhanced by considering samples from additional phases (Additional file 1: Supplementary text, Additional file 1: Figure S4d-i).

We also confirmed that the signal identified here is distinct from the signal associated with variants in promoter regions in a previous analysis of the SSC cohort [13]. Specifically, from the initial set of 12,293 M-F upstream variants, we removed those within 2 kbp upstream of its assigned TSS following the previously used promoter definition [13],

which left 11,395 variants. We then applied ENSAS and observed a similar number of significant neighborhoods before and after removing promoter variants (28 vs. 21 neighborhoods, with 12 shared neighborhoods), with a Spearman correlation of 0.92 between the two sets of *p*-values (Additional file 1: Figure S5). That we were able to observe the signal after excluding the promoter regions was expected as most variants we considered were outside of promoter regions. We also note that our analysis was not directly comparable to the previously reported promoter association analysis, since the previous promoter association analysis was based on a different analytical framework that did not directly test for sequence differences between proband and sibling variants and used a different set of de novo variant calls.

### Investigating sequence signal beyond local GC content

To first establish that ENSAS is powered to capture sequence context beyond local GC content, we simulated datasets containing variants from two groups with differences in their chromatin state distribution (Methods). We applied ENSAS to the simulated datasets and compared predictive performance using local GC content and Naive Bayes models with different *k*-mer lengths. We observed that under different simulation parameters (Methods) the best Naive Bayes models can consistently outperform local GC content (Additional file 1: Figure S6). These results confirm that a *k*-mer-based Naive Bayes model is capable of identifying association signals beyond local GC content.

We next applied ENSAS' Naive Bayes analysis to investigate if the sequence context represented by *k*-mers is more predictive of proband sibling status compared to local GC content in specific neighborhoods. We separately used *k*-mers of lengths 1 to 7 as features and for comparison purposes local GC content was also evaluated on the same testing folds. None of the *k*-mer lengths led to improved overall performance of the Naive Bayes model over local GC content (Fig. 4a). When restricted to the 28

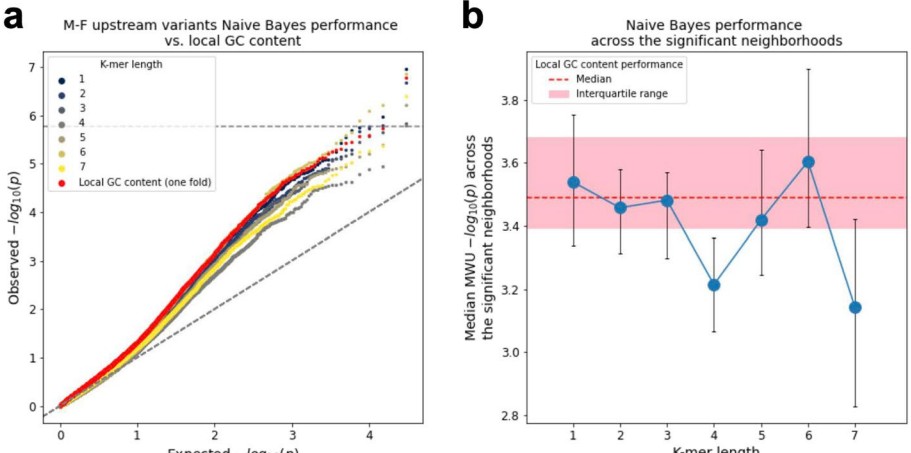

**Fig. 4** Naive Bayes model vs. local GC content predictive performance for the M-F upstream variant neighborhoods. **a** QQ-plot of the proband vs. sibling Mann–Whitney *U p*-values for the Naive Bayes scores, using *k*-mers of length 1-7 along with local GC content, all evaluated on the same testing folds. Diagonal dashed line shows the unit slope. **b** Median Mann–Whitney *U p*-values across the 28 significant neighborhoods identified based on local GC content using all variants for Naive Bayes models using *k*-mers of length 1-7, compared to the local GC content (median *p*-value shown as red dashed line, interquartile range shown as shaded area). Error bars indicate interquartile range

neighborhoods with significant proband-sibling local GC content differences previously identified using all variants, the 6-mer model showed the highest median performance, but the difference compared to local GC content did not reach statistical significance (one-sided Wilcoxon signed-rank $p = 0.34$ for 6-mer, Fig. 4b). When the random train-test splitting was repeated 100 times for each of these 28 neighborhoods, the 6-mer Naive Bayes model had better median performance than local GC content in 18 of the neighborhoods (Additional file 1: Figure S7a). We also directly tested whether individual 6-mers exhibited significant difference in prevalence between proband and sibling variants within these neighborhoods (Additional file 1: Figure S7b). We did not observe any individual 6-mer reaching statistical significance. However, as would be expected by the results with local GC content, we did observe that high-GC 6-mers are more enriched in probands while low-GC 6-mers are more enriched in siblings (Additional file 1: Figure S7c). We further observed that 6-mers containing only A's and T's had more significant $p$-values on average than sets of 6-mers with other frequencies of A's and T's (Additional file 1: Figure S7d).

Despite the results we observed in the $k$-mer Naive Bayes model and the individual 6-mer analysis, we cannot exclude the possibility that there exist additional sequence association signals beyond local GC content that could be detected with other predictive models based on $k$-mer features or other sequence representations.

### Chromatin state annotation of variants in top neighborhoods is partially predictive of proband-sibling local GC content differences

We next investigated the extent to which the proband-sibling local GC content differences in the top neighborhood (surrounding the gene OPCML) in the M-F upstream variant analysis was expected based on differences in the cell type-specific chromatin state annotations of the variants. For this we utilized annotations from a ChromHMM 18-chromatin state model defined based on six histone modifications that were available for 98-reference epigenomes from the Roadmap Epigenomics Consortium [30, 31]. For each reference epigenome, we evaluated the extent to which the differences in local GC content rank between proband and sibling variants are predicted by the variants' chromatin state annotations (Methods). Across all epigenomes, the chromatin state annotations in an individual epigenome alone predicted a median of 19.1% of the local GC content differences but reached as high as 50.2% predicted based on the annotations in a foreskin fibroblast sample. Among the epigenomes in the top 10 in terms of the amount of local GC content differences that could be predicted based on the chromatin state annotations there was representation from four different fetal samples and three fibroblast samples (34.5–50.2% predicted) (Table 1). Meanwhile, chromatin state annotations of epigenomes corresponding to brain tissues predicted a lower amount (13.6–28.3%, Additional file 1: Figure S8, Additional file 2: Table S4).

For the top 10 epigenomes whose chromatin states collectively predicted the highest percentages of the local GC content differences, we examined the contribution from each individual chromatin state (Methods). We observed that the quiescent state (*Quies*) had the largest contribution to the local GC content differences predicted, with an average of 14.7% across the 10 epigenomes. This state preferentially overlapped sibling variants compared to proband variants in all the 10 epigenomes, with siblings having 9.2%

**Table 1** Top epigenomes from Roadmap Epigenomics based on percentage of differences in local GC content between proband and sibling variants predicted by the variants' chromatin state annotations for the top neighborhood in the M-F upstream analysis

| Roadmap Epigenome ID | Description | Percentage predicted |
|---|---|---|
| E055 | Foreskin Fibroblast Primary Cells skin01 | 50.2 |
| E092 | Fetal Stomach | 48.1 |
| E017 | IMR90 fetal lung fibroblasts Cell Line | 47.1 |
| E090 | Fetal Muscle Leg | 40.6 |
| E118 | HepG2 Hepatocellular Carcinoma Cell Line | 37.3 |
| E080 | Fetal Adrenal Gland | 34.9 |
| E056 | Foreskin Fibroblast Primary Cells skin02 | 34.5 |
| E006 | H1 Derived Mesenchymal Stem Cells | 31.7 |
| E058 | Foreskin Keratinocyte Primary Cells skin03 | 29.9 |
| E034 | Primary T cells from peripheral blood | 29.7 |

| State name | Description | % GC diff. predicted | % P-S diff. | % of all variants |
|---|---|---|---|---|
| TssA | Active TSS | 0.1 | 0.0 | 0.6 |
| TssFlnk | Flanking TSS | 1.5 | 0.4 | 0.6 |
| TssFlnkU | Flanking TSS Upstream | 0.5 | 0.1 | 0.4 |
| TssFlnkD | Flanking TSS | 0.6 | 0.1 | 0.2 |
| Tx | Strong Transcription | 4.9 | 2.3 | 1.9 |
| TxWk | Weak Transcription | -0.3 | 0.6 | 4.8 |
| EnhG1 | Genic Enhancer 1 | 1.5 | 0.5 | 0.4 |
| EnhG2 | Genic Enhancer 2 | 1.4 | 0.4 | 0.2 |
| EnhA1 | Active Enhancer 1 | 1.3 | 0.5 | 0.7 |
| EnhA2 | Active Enhancer 2 | -0.2 | -0.2 | 0.6 |
| EnhWk | Weak Enhancer | 1.8 | 1.0 | 2.1 |
| ZNF/Rpts | ZNF genes & Repeats | 2.2 | 0.8 | 2.1 |
| Het | Heterochromatin | 1.0 | -0.5 | 10.3 |
| TssBiv | Bivalent/Poised TSS | 0.3 | 0.1 | 0.9 |
| EnhBiv | Bivalent Enhancer | 3.5 | 0.8 | 1.1 |
| ReprPC | Repressed PolyComb | 3.1 | 1.3 | 8.3 |
| ReprPCWk | Weak Repressed | 1.2 | 1.3 | 23.3 |
| Quies | Quiescent/Low | 14.7 | -9.2 | 42.0 |

**Fig. 5** Contributions of each chromatin state to the local GC content rank differences predicted for the top neighborhood in the M-F upstream analysis. Percentages shown are mean across the top 10 epigenomes whose chromatin states collectively explained the highest percentages of the local GC content rank differences. % GC diff. predicted: mean percentage of proband-sibling local GC content rank difference predicted based on variant annotations to this chromatin state. The smallest value is colored white and the largest value is colored purple. % P-S diff.: mean difference between proband and sibling percentages of variants overlapping the chromatin state, with positive indicating preferential overlap with probands. The largest positive value is colored blue and the largest negative value is colored red. % of all variants: mean percentage of variants in the top neighborhoods overlapping each of the states. The smallest value is colored white and the largest value is colored purple

more variants assigned to this state on average (Fig. 5, Additional file 2: Table S5, S6). We note that the large contribution from the *Quies* state was in part because it is the largest state, overlapping with 42% of all variants on average across the top 10 epigenomes. The state with the second largest contribution on average was the strong transcription state (*Tx*), which constituted 1.9% of all variants but predicted 4.9% of the local GC content differences and preferentially overlapped proband variants in all the 10 epigenomes (Fig. 5, Additional file 2: Table S5, S6). The state with the next largest contribution on average was the bivalent enhancer state (EnhBiv), constituting 1.1% of the variants while predicting 3.5% of the local GC content differences and preferentially overlapping proband variants in all 10 epigenomes (Fig. 5, Additional file 2: Table S5, S6).

### Specificity of association signal with respect to sex-upstream/downstream combinations and ASD phenotype

To determine the extent to which the association signal was unique to M-F upstream variants, we applied ENSAS to male proband-female sibling downstream (M-F downstream), male proband-male sibling upstream (M-M upstream) and male proband-male sibling downstream (M-M downstream) variants separately. We used the same set-up as the M-F upstream analysis except for the Naive Bayes analysis we restricted to $k$-mers of length 6. We observed substantially less significant $p$-values in these analyses compared to the M-F upstream analysis (Additional file 1: Figure S9). Using either Bonferroni-based or permutation-based multiple testing corrections, we did not observe any significant associations between local GC content and ASD phenotype among the three other combinations after controlling for multiple testing. The most significant $p$-values for the other combinations were $p = 8.6 \times 10^{-5}$ for M-F downstream, $2.2 \times 10^{-5}$ for M-M upstream, and $2.3 \times 10^{-4}$ for M-M downstream compared to $1.3 \times 10^{-7}$ for M-F upstream (Additional file 1: Figure S9a). The Naive Bayes analysis also did not yield any apparent signal beyond local GC content for these combinations (Additional file 1: Figure S9b). Overall, these results support the M-F and upstream specificity of the association signal.

To further investigate the specificity of the signal, we asked whether similar male–female differences are seen in an independent dataset unrelated to the ASD phenotype. For this, we applied ENSAS to de novo variants from a WGS dataset consisting of 2976 trios of Icelandic origin [32]. After applying the same filtering steps that we had applied in the SSC analysis (Methods), we applied ENSAS by comparing all male samples against all female samples and observed no neighborhoods significant for the local GC content analysis after either Bonferroni-based or permutation-based multiple testing corrections (Additional file 1: Figure S10a). The Naive Bayes analysis also did not yield any significant findings (Additional file 1: Figure S10b). This suggests that the identified male–female association signal is not common to all de novo datasets and may be related to ASD phenotypic differences, though we cannot exclude technical differences between the two cohorts.

## Discussion

Here we revisited previous findings of ASD associations identified through a deep learning-based approach [11] and showed that local GC content was sufficient to produce similar ASD associations for noncoding variants excluding CSS variants. We further showed that after conditioning on local GC content, the reported significant association signals were no longer significant. However, these analyses did not exclude there is additional sequence association signal beyond local GC content. The analyses also did not directly provide insight into why local GC content was sufficient to identify the reported associations including a suggested preferential association for variants differentially expressed in brain tissues. To gain insights into this association, we first extended the analyses to consider the sex of the probands and siblings. This suggested among male probands, which represent the large majority of probands, the signal was specific to those that have a female sibling as opposed to a male sibling. We were able to further isolate this signal to variants upstream of their assigned genes.

To more systematically investigate this association signal, we developed ENSAS. ENSAS generalizes the previous analyses by considering sequence signals that can be captured in *k*-mers instead of only local GC content. In addition, the gene expression neighborhoods it defines are designed to have greater coverage of expression space than the previously defined gene sets based on GTEx differential tissue expression. We applied ENSAS to M-F upstream variants and showed that incorporating *k*-mer-based sequence context did not lead to overall improved prediction of proband-sibling status compared to local GC content given current sample sizes. However, there still exists the possibility that sequence information beyond local GC content could identify enhanced association signals. The top neighborhoods where proband-sibling local GC content differed the most showed the strongest enrichment for synapse-related GO terms, consistent with previous findings from ASD de novo coding variants [6, 28]. We identified epigenomes whose chromatin states could collectively predict a large portion of the proband-sibling difference in the top neighborhood, primarily those pertaining to fetal and fibroblast samples. Across these epigenomes, we observed the largest contribution to the GC content predicted based on chromatin state annotations from the preferential overlap of the quiescent state among sibling variants and the preferential overlap of the strong transcription state and bivalent enhancer state among proband variants the latter of which has a strong association with developmental functions [33, 34].

Understanding why the signal was specific to male probands with female siblings could be an avenue of investigation for future studies. One hypothesis to investigate is that some of the mutations occurred post-zygotically early enough in development [35] to be found in a large fraction of cells and sex-specific chromatin or transcriptomic differences could be associated with the mutational differences [36]. Potentially related, previous research found genes with putatively damaging de novo coding variants were more highly expressed in prenatal female brains than in males, suggesting potential compensatory effects in females [36]. Also potentially relevant, another study reported elevated levels of the repressive chromatin mark H3K27me3 in female placenta compared to males, which reduced female fetuses' vulnerability to gene expression disruptions in the developing hypothalamus, a region associated with ASD [37]. Another avenue for future investigation is to understand why the observed signal is specific to variants in

regions upstream of TSS relative to downstream and the extent to which it might be related to different transcriptional processes or chromatin environments associated with these regions. We did show the association cannot be explained by variants in the immediate upstream promoter regions, for which a previous association with ASD [13] was reported. We also note that this association signal that we identified does not necessarily imply a causal relationship with ASD. It is possible for instance that this signal along with the previously established sex differences in transcription or prenatal chromatin environment is all tied to a common underlying causal mechanism.

While our analysis and ENSAS revealed findings about noncoding variant associations with ASD, we note some limitations and caveats of our analyses and results. We only applied ENSAS to de novo variants, while its applications to rare variants from non-family-based case–control studies remain an area for future investigation. However, given the weaker expected effect size of non-de novo variants this can be a challenging task, and effective application may require incorporation of stringent variant filtering thresholds such as variants not already present in aggregation databases [38]. We also note that ENSAS neighborhood strategy, while allowing systematic coverage of the expression space and removing the confounder of different sizes for variant sets, does lead to a large number of overlapping neighborhoods. While ENSAS supports both Bonferroni- and permutation-based strategies for controlling for multiple testing, future work could investigate alternative approaches to reduce the number and overlap of neighborhoods tested with ENSAS such as through clustering or selecting representative neighborhoods prior to testing.

A caveat in interpreting the results of ENSAS is that while GC content can be associated with biological features such as promoters [39], enhancers [40], gene structure [41], and mRNA decay [42], it can also correlate with sequencing technical factors. For example, sequencing read fragments can be biased towards GC-rich regions, leading to positive correlation between GC content and sequencing coverage [19]. As a result, ENSAS does not directly determine if sequence differences are due to biological differences or technical confounders associated with sequencing [19]. We note that following Zhou et al. [11], we used a repeat-masked subset of variants and thus have excluded a set of variants more vulnerable to technical confounders [18]. In addition, we showed that the association signal appears mainly driven by proband-sibling pairs with matching sequencing lane information thus suggesting sequencing differences is less likely an explanation. The specificity of the signal particularly for brain-associated genes, male probands with female siblings, and variants upstream of genes relative to downstream are also suggestive of a biological basis as it would exclude technical confounders that would not display this level of specificity. We note that even assuming the association signal is not driven by technical sequence confounders we cannot exclude the possibility that there is a biological basis for it that is correlated with the phenotype but is not directly causally related, as noted above.

We also note caveats in our analysis related to statistical considerations. A major issue in whole-genome sequencing studies is the handling of multiple tests. In our study, we employed distinct sets of tests that differed in their frameworks and test samples. We applied multiple testing corrections independently for each set instead of a uniform correction across all sets to preserve these differences, but our reported *p*-values do not

reflect multiple testing across different families of analyses. We also recognize that our analysis inherited test procedures and parameters that already generated positive findings in the previous study by Zhou et al. [11], and therefore our observed signal can be subject to inflation. Although a main finding that the brain expression signal is concentrated in a specific subset of variants (male proband-female sibling pairs, upstream of TSS) was not reported by Zhou et al. [11], we still cannot exclude the possibility that the signal is driven by technical factors and requires replication in larger samples to increase confidence in its biological significance.

## Conclusions

The role of rare noncoding variants in complex phenotypes remains an underexplored area. With the increasing availability of large-scale WGS datasets, there is a need for effective analytical frameworks. Here we presented an analytical framework that simplifies and provides new insights into the role of noncoding de novo variants in ASD and holds promise for its application in other WGS studies.

## Methods

### Defining local GC content

For each of the 127,140 de novo variants from Zhou et al. [11], which were all autosomal single-nucleotide variants not overlapping repetitive elements, we extracted the DNA sequence that was within 100 bp on each side to have sequences of length 201 bp using the GRCh37 assembly from UCSC genome browser [43] release 14. We then counted the number of nucleotides that were a "G" or "C" in the 201 bp for each sequence.

### DIS adjusted for local GC content

To construct a DNA DIS score adjusted for local GC content, for each variant we took the mean DNA DIS score of all other variants that had the exact same number of "G" or "C" nucleotides within 100 bp and then subtracted that value from the variant's DNA DIS value. We note that the DIS values globally were already mean-centered around 0. We repeated the same procedure for the RNA DIS, except restricted to the 77,157 variants that had an RNA DIS available. There were four variants for the DNA DIS that had a unique count for the number of "G" or "C" nucleotides, which were outliers, and for these variants we did not subtract any value, and similarly for three outlier variants for the RNA DIS.

### Defining coding regions and canonical splice sites

Throughout our analyses, coding regions were defined by Gencode [44] v24 annotations lifted to GRCh37, and canonical splice sites (CSS) were defined by any variants annotated as a "splice_acceptor_variant" or "splice_donor_variant" by the Ensembl Variant Effect Predictor [45] release 109.3 for GRCh37 release 98.

### Genomic variant set analysis

For computing the significance of proband and sibling score differences for different sets of all genomic variants, we followed the same procedures as used by Zhou et al. [11]. We used the version of the variant annotations used by Zhou et al. [11], which was provided

by them. We note there are small differences between that version and the version of the annotations provided in Supplementary Table 1 of Zhou et al. [11] for the RNA gene sets, due to differences in the gene identifiers used.

We tested the 130 variant sets using (1) local GC content, (2) the original DNA and RNA DIS reported (60 sets were tested with DNA DIS, 70 sets were tested with RNA DIS), and (3) the DNA and RNA DIS adjusted for local GC content, described above. The $p$-value significance of proband mutations was computed with a one-sided Mann–Whitney $U$ test and false discovery rates were computed with the Benjamini–Hochberg procedure. We also conducted an additional set of tests for each score where we first removed any variant in coding regions or CSS from the variant set and then applied the same procedures.

### Tissue-specific gene expression analysis

We analyzed tissue-specific gene expression from GTEx following the procedures of Zhou et al. [11]. Each of the variants was assigned to their respective genes based on the distance to a nearest representative TSS as described by Zhou et al. [11]. We used the list of GTEx tissue-specific genes for each of the 53 GTEx tissues provided by Zhou et al. [11] (Additional file 3), which was defined as those with five times its median expression across all tissues. We note that for this analysis the authors created a single combined score of the RNA and DNA DIS. We followed the author's procedure in their provided code for creating the combined score. Specifically in this analysis, if a variant had an RNA DIS value available, the mean of the DNA and RNA DIS was used; otherwise, just the DNA DIS was used. Also following the code provided by the authors, only variants within 100 kbp of a TSS or intron variants within 400 bp of an exon boundary were included.

For each of the GTEx tissues, we performed a one-sided Mann–Whitney $U$ test on the local GC content of proband variants versus sibling variants assigned to these genes. We repeated the same test using DIS scores and DIS scores corrected for local GC content, and with coding and CSS variants removed. The multiple testing correction was done with the Benjamini–Hochberg procedure.

To separately test variants upstream of the outermost TSS of their assigned gene from those downstream, we remapped the assignment of variants to genes. For each gene, we determined the position of the outermost TSS, which was the lowest coordinate if the gene was on the positive strand and the greatest coordinate if the gene was on the negative strand. We removed variants overlapping the same set of annotations used for defining coding regions and CSS as described above. We then assigned each remaining variant to its nearest outermost TSS. Variants with a coordinate less than the TSS of its assigned gene where the gene is on the positive strand or with a coordinate greater than the TSS of its assigned gene where the gene is on the negative strand were classified as upstream variants. The remaining variants were classified as downstream variants.

We then repeated the above testing procedure separately using different subsets of variants:

(A) Variants from proband-sibling pairs where the proband is male and the sibling is female.

(B) Variants from proband-sibling pairs where the proband is male and the sibling is male.

(C) Same proband-sibling sex combination as (A) but the variant is within 100 kbp upstream of the nearest TSS.

(D) Same proband-sibling sex combination as (A) but the variant is within 100 kbp downstream of the nearest outermost TSS.

For (A), (B), (C), and (D), we removed coding and CSS variants. For (C) and (D), we did not include variants based on proximity to exon boundaries. Additional analyses for other subsets of variants are described in Additional File 1: Supplementary text.

### Defining PTVs

We annotated the de novo variants using the Ensembl Variant Effect Predictor for GRCh37. We defined PTVs to be those annotated as "stop_gained," "splice_donor_variant," "splice_acceptor_variant," or "frameshift_variant," following the definitions previously used by Fu et al. [10].

### Enrichment tests for de novo coding variant and PTV counts

For each sample, we counted the total number of de novo variants. Following Werling et al. [12] and An et al. [13], we adjusted the de novo variant counts based on paternal age. We performed a linear regression using the paternal age of each sample as the predictor variable and the total number of de novo variants of each sample as the response variable. The residual of the regression model was then shifted such that its sample-wise mean is the same as the mean number of de novo variants across samples before adjustment. This shifted residual was taken as the new adjusted counts. For each sample, we computed the ratio between adjusted and raw counts. To get the adjusted counts of de novo coding variants and PTVs, their respective raw counts were multiplied by this sample-specific ratio. We used two-sided binomial tests to separately test for proband-sibling differences in de novo coding variant and PTV counts.

### Defining gene expression neighborhoods

To define gene expression neighborhoods, we first obtained the pairwise gene–gene co-expression correlation matrix from Geneshot [22], which is computed using the RNA-seq data compiled by the ARCHS4 resource [23]. Based on statistics from the current version of ARCHS4, for which sample composite statistics is available, though we note this is a newer version than we used through the Geneshot database, we estimate 2.5% of the samples being brain-related. These brain-related samples comprise various brain regions such as midbrain, prefrontal cortex, and others. Of these brain-related samples, we estimate 2.6% of them being fetal brain samples. We estimate 20.5% of the total samples have sex labels, with 68% of these samples labeled as male and 32% as female. The correlation matrix contained a total of 29,820 genes. We defined the "neighborhood" $N_g$ surrounding each gene $g$ to be the top $M$ variants that are assigned to its closest genes

based on the pairwise distances, including the variants assigned to $g$ itself, with ties broken arbitrarily. This strategy is similar to the gene set augmentation functionality of Geneshot [22]. We also greedily pruned variants that are $< L/2$ bp apart, that is for each pair of variants in the input with $< L/2$ bp between each other we kept the variant with the smaller coordinate, where $L$ is the length of the sequence centered at each variant in ENSAS.

### ENSAS: Expression Neighborhood Sequence Association Study

To test for the local GC content differences between two groups of variants in a neighborhood, ENSAS uses a Mann–Whitney $U$ test which could be either one-sided or two-sided depending on the specific application.

To test for higher-order sequence differences between the two groups, ENSAS uses a machine-learning-based approach. For each variant in a neighborhood, ENSAS extracts the $L$ bp nucleotide sequence centered around it. Then for a given $k$, it counts the number of occurrences of each $k$-mer in the sequence around each variant. For each neighborhood, the variants are split into evenly sized training and testing folds. A multinomial Naive Bayes classifier is trained on the training fold to distinguish between variants in the two groups, using each variant's $k$-mer occurrence counts as features and uniform class priors. One of the groups is labeled as positive and the other group is labeled as negative. The classifier is then applied to the testing fold to compute the posterior probability ("score") of each testing variant being in the positive group. ENSAS then tests the difference in the distribution of scores between the two groups of variants in the testing fold with a one-sided Mann–Whitney $U$ test, with the null hypothesis being that the variants with positive labels have overall higher scores. The same test is also performed using local GC content on the testing fold. ENSAS by default computes a Bonferroni-corrected $p$-value significance threshold of $0.05/n$ where $n$ is the number of neighborhoods. As described below ENSAS, also provides a permutation based multiple testing correction option.

We applied ENSAS to variants upstream of TSS in male proband-female sibling pairs. Given the high correlation of local GC content and DIS, for this application we used one-sided tests for local GC content differences with probands and siblings labeled positive and negative, respectively, to maintain consistency with the use of one-sided tests for DIS in Zhou et al. [11]. Across the GTEx tissues, the mean number of variants (within the male proband-female sibling upstream subset) whose genes show differential expression was 1048. We therefore set the size of neighborhoods $M$ to be 1000 to approximate that. We set the sequence length $L$ to be 201. We separately tested 1,2,3,4,5,6 and 7-mers. For the other sex-upstream/downstream combinations (male proband-female sibling downstream, male proband-male sibling upstream, male proband-male sibling downstream), we also performed ENSAS with neighborhood sizes of 1000 and a $k$-mer length of 6.

To investigate the significance of individual $k$-mers for each of the top 28 neighborhoods with significant proband-sibling local GC content differences, within each neighborhood for each 6-mer we used a two-sided binomial test to test if the 6-mer is significantly more prevalent in the proband variants or the sibling variants. We controlled for multiple testing using a Bonferroni-corrected $p$-value threshold of $0.05/4^6$.

### Permutation-based multiple testing correction

In addition to the Bonferroni-based correction, ENSAS also provides a permutation-based approach to correct for multiple testing. Let the Mann–Whitney $U$ $p$-value of a neighborhood $i$ on the actual data be $p_i$. The number of discoveries claimed as true positives based on a $p$-value threshold $T$ is then $|\{j: p_j < T\}|$ that is the number of neighborhoods with a smaller $p$-value. In each permutation ENSAS randomly swaps the proband-sibling labels within each pair. The estimated number of false positive discoveries based on $B$ (1000 by default) permutations is $\Sigma_B|\{j: p_j^b < T\}|/B$ that is the average number of neighborhoods with smaller $p$-value than $T$ across the $B$ permutations, where $p_j^b$ is the $p$-value of neighborhood $j$ in the $b$th permutation. The estimated FDR based on threshold $T$ is the estimated number of false positive discoveries divided by the number of discoveries claimed as true positives. ENSAS finds for $T$ the largest $p_i$ such that the estimated FDR is 0.05 and declares all neighborhoods with $p$-value smaller than $p_i$ as significant.

### Gene ontology enrichment analysis for the top neighborhoods

We performed Gene Ontology (GO) enrichment analyses for the top neighborhoods with the most significant local GC content $p$-values from the ENSAS performed on M-F upstream variants. For the GO analyses, we tested each neighborhood separately. We used as the foreground the set of genes with assigned variants in each neighborhood. We separately used two different background sets of genes: (1) the union of all genes assigned to the M-F upstream variants and (2) the union of genes assigned to the M-F upstream variants that were also considered differentially expressed within any of the 13 brain-related GTEx tissues [11]. The analysis was done using GOATOOLS [46], with Fisher's exact test conducted for each GO term and FDR-controlling Benjamini–Hochberg procedure to correct for multiple testing.

### Clustering of the top neighborhoods using DBSCAN

We clustered the top 28 neighborhoods using DBSCAN [29] implemented by Scikit-learn version 0.24.1 [47], with the distance metric being the Manhattan distance between the variants of each pair of neighborhoods when encoded as binary vectors, which is equal to the neighborhood size $M$ (1000) minus the number of variants shared by the two neighborhoods. Each cluster was allowed to have as few as one neighborhood, and neighborhoods assigned to the same cluster were required to have a pairwise distance less than 500, corresponding to half of the neighborhood size, to all other neighborhoods in the same cluster.

### Stratification by matching sequencing lanes

We obtained the sequencing lane information for each sample, represented by columns "LANE_from_ID" and "LANE_from_PU." We defined proband-sibling pairs with matching sequencing lanes to be the pairs where the proband and the sibling samples have exactly the same lane assignments in both columns, and otherwise mismatching lanes. Under this criteria, among male proband-female sibling pairs there were 968 samples with matching sequencing lanes and 606 samples with mismatching sequencing lanes. Next, for each neighborhood when conducting ENSAS on the M-F upstream variants,

we performed proband vs. sibling Mann–Whitney $U$ tests separately on the subsets of variants from samples with matching sequencing lanes and samples with mismatching sequencing lanes.

### Analysis excluding variants in promoter regions

From the 12,293 M-F upstream variants, we removed variants in promoter regions, defined as < 2 kbp upstream of their assigned nearest outermost TSS following the promoter definition from An et al. [13]. We then performed ENSAS on the remaining 11,395 M-F upstream variants with a neighborhood size of 1000.

### Evaluating ENSAS with simulations

We investigated if ENSAS can differentiate between proband and sibling variants simulated to have a different underlying chromatin state distribution. To do this, we used chromatin state annotations produced by an 18-state ChromHMM model from the Roadmap Epigenomics Consortium [30, 31]. We selected an epigenome (E003-H1 cell line) and simulated 500 proband and 500 sibling labeled variants by randomly drawing positions (excluding assembly gaps) belonging to different chromatin states. Among proband variants, $X$% were drawn from the four active TSS-associated states (*TssA*, *TssFlnk*, *TssFlnkU*, and *TssFlnkD*) and the other (100-$X$)% were uniformly drawn across the genome, where $X$ was a parameter we varied. All sibling variants were uniformly drawn across the genome. We greedily pruned variants as above such that no two variants were < 100 bp apart to ensure no overlap of sequences. For each $X$ in {10, 20, 50, 80, 100}, we generated 50 simulated datasets. We applied ENSAS to distinguish between the proband and sibling variants, with a sequence length $L$ of 201 bp and a neighborhood size $M$ of 1000. We used values from 1 to 7 as candidate lengths of $k$-mers and repeated the train-test split 50 times for each simulated dataset.

### Prediction of proband-sibling sequence differences with chromatin states in the top neighborhood

For the top neighborhood identified when applying ENSAS to M-F upstream variants, we quantified the extent by which chromatin state annotations from specific epigenomes can predict the proband-sibling differences in the local GC content. We annotated each of the 1000 variants using the 18-state ChromHMM model defined based on six histone modifications across 98-reference epigenomes by the Roadmap Epigenomics Consortium [30, 31]. For each variant, we converted its local GC content to a rank value and broke ties arbitrarily, as the significance tests ENSAS applied were rank-based. We then randomly partitioned the variants into five subsets. For each variant, we predicted its rank based on the mean rank of variants from the other four subsets to which the variant belonged. For this prediction, we used variants assigned to the same chromatin state regardless of whether the variant was from a proband or sibling. If the state was not present in the training set, then the mean rank of all variants was assigned as the prediction. We took the difference between the proband mean predictions and the sibling mean predictions and then divided this difference

by the observed mean difference in rank to determine the percent of rank differences predicted by the chromatin state annotations in the reference epigenome.

To quantify the contribution of individual chromatin states to the mean prediction rank differences between proband and siblings, we computed the quantity $((a_{proband,s}-r) \times f_{proband,s}-(a_{sibling,s}-r) \times f_{sibling,s})$. $a_{proband,s}$ and $a_{sibling,s}$ are the mean rank predictions for variants in state $s$ from proband and siblings, respectively. $f_{proband,s}$ and $f_{sibling,s}$ are the fraction of predictions for variants in state $s$ from proband and siblings, respectively. $r$ is the overall mean rank of all variants. We then divided this quantity by the observed mean difference in rank and multiplied this value by 100%. We note that this value can be negative for some states.

### Application to an Icelandic dataset

We applied ENSAS to a dataset of an Icelandic population consisting of 194,687 autosomal de novo variants from 2976 WGS trios reported by Halldorsson et al. [32]. We used UCSC LiftOver to lift the variants from GRCh38 to GRCh37 and performed comparison between male samples and female samples. For the local GC content analysis, we tested whether variants from male samples had higher local GC content than those from female samples using a one-sided test. We restricted the analysis to single nucleotide variants and followed the same filtering procedures that we had applied to the SSC variants, to remove variants in coding regions and CSS or not within 100 kbp upstream of their nearest outermost TSS. We then greedily pruned variants that were < 100 bp apart. We applied ENSAS with 1000 neighbors to the remaining 41,846 variants.

### Supplementary Information

---

Additional file 1. Supplementary text, Supplementary Figures S1-S10, Supplementary Tables S2 and S3.

Additional file 2. Supplementary Tables S1, S4, S5 and S6.

Additional file 3. List of SSC *de novo* variants used in analyses and their processed information all based on the GRCh37 assembly.

Additional file 4. Peer review history.

---

#### Acknowledgements

We acknowledge the contributions of participants in the SSC and Iceland cohorts. We thank Jian Zhou, Christopher Park, Chandra Theesfeld, and Olga Troyanskaya for discussions and sharing code and data. We thank Daniel Geschwind and Stephan Sanders for discussions and comments on a preliminary draft. We thank Shan Dong and Stephan Sanders for providing annotations of the inferred sex of individuals in SSC data and phase and sequencing lane information. We thank Bjarni V. Halldórsson for providing annotations of the inferred sex of individuals in the Iceland cohort. We thank members of the Ernst lab and whole-genome sequencing for psychiatric disorders (WGSPD) consortium for discussions.

#### Review history

The review history is available as Additional file 4.

#### Peer review information

#### Authors' contributions

R.L and J.E conceived and developed the methods, performed analyses, and wrote the manuscript.

#### Funding

We acknowledge funding from US National Institutes of Health grants DP1DA044371, U01MH105578, R01MH109912, R01MH110927, U01HG012079, U01MH130995, and UCLA Jonsson Comprehensive Cancer Center and Eli and Edythe Broad Center of Regenerative Medicine and Stem Cell Research Ablon Scholars Program (J.E.) and a UCLA dissertation year fellowship (R.L.).

### Data availability

De novo variant calls from the SSC WGS data were obtained from Ref. [11]. De novo variant calls from the WGS data for the Icelandic population were obtained from Ref. [32]. Gene definitions were obtained from Gencode v24 lifted to GRCh37: https://www.gencodegenes.org/human/release_24lift37.html. Ensembl release 109.3 variant annotations were obtained from https://github.com/Ensembl/ensembl-vep/tree/release/109.3. Geneshot gene expression correlation matrix were obtained from https://maayanlab.cloud/geneshot/download.html. 18-state ChromHMM annotations for 98 epigenomes were obtained from the Roadmap Epigenomics Project: https://egg2.wustl.edu/roadmap/data/byFileType/chromhmmSegmentations/ChmmModels/core_K27ac/jointModel/final/. Source code for ENSAS under the MIT license can be found at https://github.com/ernstlab/ENSAS [48] and [49].

## Declarations

### Ethics approval and consent to participate
Not applicable.

### Consent for publication
Not applicable.

### Competing interests
The authors declare that they have no competing interests.

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
