## [Additional file 4. Peer review history. · Genome Biology]

Review history

First round of review

Reviewer 1

This work describes the development and application of the Expression Neighborhood Sequence Association Study (ENSAS) analytical framework for identification of sets of noncoding variants associated with phenotypes, applied in this case to autism spectrum disorder (ASD). Whole-genome sequencing has great promise to expand genetic diagnoses and further understanding of the genetic architecture of conditions like ASD, but interpretation of noncoding variants, and identification of individual or functionally grouped sets of variants associated with a trait has continued to prove challenging, whether by using manually defined variant sets of interest (e.g. CWAS) or by using machine learning approaches (Zhou et al 2019; DIS scores). Here, Li and Ernst simplify prior machine-learning based work by demonstrating a strong association between GC content and DIS scores, and by extending variant set association tests in an interpretable way by utilizing gene co-expression networks (here, "neighborhoods"), GC content, and k-mer based sequence content. Using the ENSAS approach, the authors identify Bonferroni-significant associations for 28 neighborhoods of synapse-annotated genes, a signal that is predominantly driven by male proband-female sibling pairs and by variants upstream of transcription start sites. This method presents a valuable tool for extending our understanding of noncoding variant associations. I do have several questions about the approach and findings:

Major comments

(1) The finding that male proband-female sibling pairs drive most of the signal is intriguing. One concern could be that the male-female pairs are better powered for discovery than male-male pairs. Though the text notes similar numbers of de novo variants in male-female and male-male pairs (30,430 and 27,251 variants), what is the breakdown of the number of sibling pairs with each configuration? Is there a significant difference in the number of pairs, or number of variants per pair?

Similarly, is there any signal to be found when considering sibling pairs with female probands? I understand that numbers will be much smaller, and likely underpowered. However, this should also be run, or at a minimum, the rationale for excluding this analysis should be directly addressed.

(2) I appreciate the analysis of Icelandic trios to compare males to females, as one possible explanation for the above result is that there is a baseline sex difference in de novo variant distribution. Could a similar analysis be run in the SSC data, comparing male and female siblings to each other, and possibly also comparing male and female probands? Though a sex difference may not be "common to all de novo data sets" as stated, it may still be a feature of the SSC cohort.

(3) The use of ARCHS4 over GTEx is a valuable extension, and more information about the contents of this database would be appreciated. Which brain tissues or cells are included? GTEx is comprised of older donors, but for a developmental disorder like ASD, gene expression networks from fetal brain may be more relevant. Does ARCHS4 include fetal brain? What is the representation of male and female samples in ARCHS4? (e.g. GTEx and many other post mortem data sets skew male)

(4) What is the overlap in gene content and variants between the 28 Bonferroni-significant neighborhoods? Do many of these neighborhoods overlap in some capacity, and/or do these 28 neighborhoods capture distinct subsets of variants? Can neighborhood signals be clustered in order to highlight associations coming from distinct gene sets/functional pathways?

(5) The test for signal differences between sibling pairs sequenced in the same vs. different lanes is an appreciated QC check, though I am surprised by the lack of signal from the lane-mismatched pairs (if sequencing lane is not a factor, one might expect similar performance from both sets). What is the median p-value for the top 28 neighborhoods in the lane-mismatched sets? Are there any known (metadata) factors associated with whether probands and siblings were sequenced in the same, or different, lanes?

(6) The ENSAS analysis run excluding putative promoter variants is interesting, and demonstrates ENSAS' ability to discover new associations. But I am also curious, how does ENSAS perform for just the putative promoter variants? Does the ENSAS approach also identify an association signal for these variants? (validation of An et al?)

(7) What is the rationale for testing up to 7 bp k-mers? Is a 7 bp k-mer long enough to capture relevant sequence context, e.g. functional elements like transcription factor binding sites? Also, are these specific k-mers sequences that are most strongly associated with the association signals?

(8) Page 23, lines 13-20 proposes a hypothesis that chromatin or transcriptomic differences between males and females may account for the increased signal in male proband-female sibling pairs. Can the authors clarify how such a mechanism could work? Is the implication that variants in the same neighborhoods in males and females may have different expression profiles, and therefore different effect sizes in males and females?

Minor comments

(1) Fig 1a) shows that DNA DIS increases as GC content increases, though the x-axis "GC content bins" is not intuitive to interpret. Can this be converted to % GC content, or perhaps a second x-axis added to show % GC content? (same for Supp Fig 1a)

Fig 1b) x-axis is labeled as "DIS -log₁₀(p)" but figure legend notes that x-axis is the proband - sibling difference in DIS. Please re-label the x-axis to clarify this.

Fig 1b) typo in legend text, repeated text "proband sibling differences"

(2) The finding that local GC content can identify noncoding association signals is interesting, and the authors note in Results that "these analyses also do not explain why local GC content, which can be both biologically significant and a confounded in genomic analyses, was sufficient to identify the association signals". If space allows, can the authors expand on these possible mechanisms, biological and technical, in the Discussion?

(3) Page 14, paragraph ~lines 14-35 describes GO enrichment analysis for the 28 neighborhoods, but the approach is confusing as described. Can the authors clarify, were the genes in each neighborhood tested separately for GO enrichment? Or was the union of genes across all neighborhoods tested for enrichment?

(4) Page 20, line 58 "we observed substantially reduced p-values" - should this read "increased" p-values (less significant)?

Reviewer 2

The topic of the effect of non-coding de novo mutations in ASD has been hotly debated, fueled by the preponderance of non-coding variants among common variants influencing heritability and by the unambiguous effect of coding de novo mutations. The community has not settled on whether non-coding de novo mutations are also a significant contributor to the ASD risk (although the question may be soon resolved with the widely anticipated new sequencing data). This manuscript has three components.

First, the authors demonstrate that previously reported apparently positive findings facilitated by a ML method can be replicated (and even enhanced) by substituting the ML machinery by a measure of GC content. Although this result is not glorious, I find it important. It reminds us that in some cases a ML black box lends itself to a simple interpretation.

Second, it is observed that the effect is limited to male-female proband-sibling pairs and to the mutations upstream of genes. These are highly surprising observations (especially, the former). It goes against the naïve expectation.

Third, the authors developed the ENSAS software. Even though it is an interesting development (and perhaps is where the larger part of the work went), it is probably the least impactful aspect of the paper. Application of ENSAS have not led to new observations, and it is not very likely that the method will be widely adopted by the field.

My specific comments are listed below.

1. The manuscript reports many p-values corresponding to numerous statistical analyses. The major critique of the field stems from the multitude of tests including definitions of regulatory regions, loci etc. It would be great seeing one flagship number, e.g. p-value corrected for all tests performed in this work.

2. I do not doubt the statistical discipline in this work. However, it inherits a lot of ad hoc parameter choices from previous studies that may have (unintentionally) lead to the inflation of the signal (at the end only positive findings out of many attempts are published). Some discussion of this would be warranted, especially given that we anticipate a new data release potentially resolving the whole issue.

3. I do not fully understand the choice of the test (Mann-Whitney on GC content). Is the expectation that a single large effect de novo mutation contributes to a subset of cases? The manuscript would benefit from a simulation (or an analysis) to show that the observed signal is in basic agreement with the known epidemiology of ASD and that the selected test is appropriate.

4. The observations that the signal is limited to the sequence upstream of TSS (but not promoter!) and to the M-F comparisons are highly intriguing (assuming they are not statistical artifacts). The difference between the M-M and M-F comparisons needs more discussions. I see two possible directions of inquiry (there are probably more that I do not see). One is that some of the mutations are post-zygotic, and there is a bias in somatic mutagenesis between males and females. This is discussed in the manuscript. This explanation is inconsistent with the analysis of the deCODE data. The other possibility is that due to some protective effect in females, brothers of probands are more likely to be affected than sisters. This would mean that the polygenic risk in probands with an unaffected sister is, on average, larger than in probands with an unaffected brother (the effect is probably small and may lead to the opposite direction of the effect but may still be explored).

5. Unless I missed it in the manuscript, is there any difference between M-M and M-F pairs in the prevalence of coding de novos?

Authors' response to reviewers

Reviewer #1: This work describes the development and application of the Expression Neighborhood Sequence Association Study (ENSAS) analytical framework for identification of sets of noncoding variants associated with phenotypes, applied in this case to autism spectrum disorder (ASD). Whole-genome sequencing has great promise to expand genetic diagnoses and further understanding of the genetic architecture of conditions like ASD, but interpretation of noncoding variants, and identification of individual or functionally grouped sets of variants associated with a trait has continued to prove challenging, whether by using manually defined variant sets of interest (e.g. CWAS) or by using machine learning approaches (Zhou et al 2019; DIS scores). Here, Li and Ernst simplify prior machine-learning based work by demonstrating a strong association between GC content and DIS scores, and by extending variant set association tests in an interpretable way by utilizing gene co-expression networks (here, "neighborhoods"), GC content, and k-mer based sequence content. Using the ENSAS approach, the authors identify Bonferroni-significant associations for 28 neighborhoods of synapse-annotated genes, a signal that is predominantly driven by male proband-female sibling pairs and by variants upstream of transcription start sites. This method presents a valuable tool for extending our understanding of noncoding variant associations. I do have several questions about the approach and findings:

Response: We thank the reviewer for the summary and positive comments. We also thank the reviewer for the constructive comments, which has led to a strengthened manuscript. We provide our responses to the comments below.

Major comments

(1) *The finding that male proband-female sibling pairs drive most of the signal is intriguing.*

One concern could be that the male-female pairs are better powered for discovery than male-male pairs. Though the text notes similar numbers of de novo variants in male-female and male-male pairs (30,430 and 27,251 variants), what is the breakdown of the number of sibling pairs

Response: We thank the reviewer for the questions. There are 787 male proband-female sibling pairs and 713 male proband-male sibling pairs. There is no significant difference between the mean number variants per individual between the two configurations (two-sided Mann-Whitney U p-value=0.4). We added the following sentence in the "Local GC content differences around brain-specific genes are specific to male probands with female siblings" results section to reflect this information:

Specifically, we repeated the gene expression association tests based on local GC content separately for de novo variants from a family with a male proband and a female sibling (n=30,430 from 787 families) and for variants from a family with a male proband and a male sibling (n=27,251 from 713 families). We note there was no significant difference between the number of de novo variants per sample between the two groups (two-sided Mann-Whitney U p-value=0.4; average 19.33 variants for samples from male proband-female sibling pairs and 19.11 variants for samples from male proband-male sibling pairs). We note that consistent with

the previous analysis all variants that we considered were on autosomes and excluded those variants overlapping repeats

In addition, we have now added an analysis where we downsampled the male proband-female sibling variants to 27,251, matching the number of male proband-male sibling variants. We observed relatively similar levels of significance compared to using all male proband-female sibling variants. We added this analysis to the “*Local GC content differences around brain-specific genes are specific to male probands with female siblings*” results section on page 9:

To confirm that the differences between the male proband-female sibling and the male proband-male sibling were not driven by the modest sample size differences, we downsampled the male proband-female sibling variants to 27,251, matching the count of male proband-male sibling variants. This random downsampling was performed 10 times, followed by a repeated analysis. We still observed 11 out of 13 brain-related tissues with significant median p-values, with the most significant tissue having a median p-value of 0.00031 (Supplementary Figure 1e).

We also updated Supplementary Figure 1 to include a figure for this analysis (Supplementary Figure 1e, next page)

(1, continued) Similarly, is there any signal to be found when considering sibling pairs with female probands? I understand that numbers will be much smaller, and likely underpowered. However, this should also be run, or at a minimum, the rationale for excluding this analysis should be directly addressed.

Response: We thank the reviewer for the comment. We conducted additional analysis to test female proband-male sibling pairs (3,948 variants from 101 families) and female proband-female sibling pairs (4,664 variants from 123 families). We did not observe a significant signal, but as the reviewer notes this analysis is likely underpowered. We added this analysis to the “*Local GC content differences around brain-specific genes are specific to male probands with female siblings*” results section:

We repeated the analysis for *de novo* variants from a family with a female proband and a female sibling (n=4,664) and for variants from a family with a female proband and a male sibling (n=3,948), and in both cases we did not observe any significant signal though we note the much smaller sample size for this analysis (Supplementary Figure 1d).

We also updated Supplementary Figure 1 to include a figure for this analysis (Supplementary Figure 1c, next page)

Supplementary Figure 1 DIS vs. local GC content. Additional panels related to Fig. 1. **(a)** The y-axis corresponds to the DNA DIS score and the x-axis evenly sized local GC content intervals, each with size 10 (corresponding to 5%, with bin labeled as N representing N%-5% - N%). The boxes correspond to the quartiles, the lengths of whiskers correspond to 1.5x interquartile range and variants above/below the whiskers are defined as outliers. Spearman correlations are computed between DIS and local GC content, across all variants. **(b-h)** Proband-sibling differences for DIS (combined) or local GC content of variants assigned to each of the 53 GTEx tissue or cell types are shown, with coding and canonical splice site (CSS) variants removed.

Both the x- and y-axis show $-\log_{10}$ p-values from one-sided Mann-Whitney U-tests. Horizontal and vertical dashed lines show p-values at FDR threshold of 0.05. Points greater than (but not on) these lines are significant after FDR correction. Diagonal dashed lines show the unit slope. **(b)** Proband-sibling differences for DIS (as conducted by Zhou et al.¹) vs. local GC content; **(c)** Proband-sibling differences for DIS (adjusted for local GC content) vs. proband-sibling differences for DIS. **(d)** Female proband-male sibling pair differences for local GC content vs. female proband-female sibling pair differences for local GC content. **(e)** Male proband-female sibling pair differences for local GC content, in a downsampled subset of 27,251 variants to match the number of male proband-male sibling variants, vs. male proband-male sibling pair differences for local GC content. **(f)** All male siblings-all female siblings differences vs. all male proband-all female proband differences for local GC content. **(g)** Male siblings (with male probands)-female siblings (with male probands) vs. male probands (with female siblings)-male probands (with male siblings) differences for local GC content. **(h)** male proband-female sibling pair differences in variants <100kbp upstream of nearest outermost TSS only for local GC content vs. for DIS.

(2) *I appreciate the analysis of Icelandic trios to compare males to females, as one possible explanation for the above result is that there is a baseline sex difference in de novo variant distribution. Could a similar analysis be run in the SSC data, comparing male and female siblings to each other, and possibly also comparing male and female probands? Though a sex difference may not be "common to all de novo data sets" as stated, it may still be a feature of the SSC cohort.*

Response: We thank the reviewer for the comment. We have now added male sibling-female sibling and male proband-female proband comparisons. While no tissues were significant at an FDR threshold of 0.05, we observed 10 and 1 out of 13 brain-related tissues being nominally significant ($p < 0.05$) for the male sibling-female sibling and male proband-female proband comparisons respectively. We updated Supplementary Figure 1 to include a figure for this analysis (Supplementary Figure 1f, above) and included these results as the section "Local GC content differences between male and female samples" in the Supplementary Text:

To assess the extent to which the observed signal might be associated with general differences between male and female samples, we repeated the analysis by comparing male siblings with female siblings, as well as comparing male probands with female probands. For both comparisons, no tissue reached the FDR-based significance threshold of 0.05. Between variants from male siblings ($n=15,685$) and female siblings ($n=17,272$) we note that 10 out of the 13 brain-related tissues are nominally significant ($p < 0.05$), (Supplementary Figure 1f). Between variants from male probands ($n=30,129$) and female probands ($n=4,496$) one brain tissue was nominally significant ($p < 0.05$, Supplementary Figure 1f) though we note the sample size of variants for female probands is limited in this analysis. These results provide suggestive evidence of differences between male siblings and female siblings though it did not reach the same level shown by the male proband-female sibling analysis.

We also note that siblings in the SSC dataset are not representative of unaffected samples in the general population as the Icelandic cohort is because they were conditioned on coming from families with predisposed ASD risk (as they already have an affected brother or sister). The male-female sibling difference can thus still reflect the different ASD risk factors in the families. We included this reasoning in the same section:

We emphasize that the SSC dataset only consists of samples from ASD families and thus siblings are not necessarily representative of unaffected samples in the general population.

To examine the impact of the male-female sibling difference in the previous male proband-female sibling vs. male proband-male sibling analysis, we also restricted the male sibling-female sibling comparison to those in families with a male proband. In this analysis, no tissues were significant at an FDR threshold of 0.05 but 9 out of the 13 brain-related tissues are nominally significant. We updated Supplementary Figure 1 to include a figure for this analysis (Supplementary Figure 1g, above) and updated the same section:

To further explore the potential sex differences among siblings, we restricted the analysis to siblings from families with a male proband (13,571 male sibling variants and 14,891 female sibling variants, Supplementary Figure 1g). While no significant signals were found at an FDR threshold of 0.05, 9 out of the 13 brain-related tissues are nominally significant.

Additionally we compared the proband counterparts of the above analysis, that is between male probands with female siblings and male probands with male siblings. In this analysis, no tissues were significant at an FDR threshold of 0.05 and two brain-related tissues are nominally significant. We updated Supplementary Figure 1 to include a figure for this analysis (Supplementary Figure 1g, above) and updated the same section:

We also compared variants from male probands with female siblings ($n=15,539$) and male siblings ($n=13,680$), observing no significant signal at an FDR threshold of 0.05 though two brain-related tissues were nominally significant ($p < 0.05$).

Overall, these results suggest that the signal originally observed between male proband-female siblings but not male proband-male siblings was driven by a contribution from both the difference between male and female probands with female siblings and the difference between male and female siblings with male probands. We updated the same section to mention this:

These results suggest the signal originally observed between male proband-female siblings but not male proband-male siblings is not explained by only the difference between male probands with male siblings and male probands with female siblings or the difference between male and female siblings with male probands.

All these results are referred to by the following paragraph added to the “*Local GC content differences around brain-specific genes are specific to male probands with female siblings*” results section in the main manuscript:

To assess the extent to which the observed signal might be associated with general differences between male and female samples, we repeated the analysis by comparing male siblings with female siblings, as well as comparing male probands with female probands. We did not observe any significant signal after multiple testing corrections, though some associations were nominally significant (Supplementary Text, Supplementary Figure 1f). Comparing male probands with female siblings and male probands with male siblings also did not yield any significant associations after multiple testing corrections (Supplementary Text, Supplementary Figure 1g).

(3) *The use of ARCHS4 over GTEx is a valuable extension, and more information about the contents of this database would be appreciated. Which brain tissues or cells are included? GTEx is comprised of older donors, but for a developmental disorder like ASD, gene expression networks from fetal brain may be more relevant. Does ARCHS4 include fetal brain? What is the representation of male and female samples in ARCHS4? (e.g. GTEx and many other post mortem data sets skew male)*

Response: We thank the reviewer for the questions. We estimated these values based on the current version of the ARCHS4 for which this information is available, but note that the version of the ARCHS4 data behind the Geneshot gene correlation used in our analysis was a prior version. Based on statistics from the current version of ARCHS4, we estimate 2.5% of the samples being brain-related. These brain samples comprise various brain regions such as midbrain, prefrontal cortex and others. Of these brain-related samples, we estimate 2.6% of them being fetal brain samples. Based on querying the current ARCHS4 database we estimate 20.5% of the total samples have sex labels, with 68% of these samples labeled as male and 32% as female. We included this information in the “Defining gene expression neighborhoods” methods section:

Based on statistics from the current version of ARCHS4, for which sample composite statistics is available though we note this is a newer version than we used through the Geneshot database, we estimate 2.5% of the samples being brain-related. These brain-related samples comprise various brain regions such as midbrain, prefrontal cortex and others. Of these brain-related samples, we estimate 2.6% of them being fetal brain samples. We estimate 20.5% of the total samples have sex labels, with 68% of these samples labeled as male and 32% as female.

(4) *What is the overlap in gene content and variants between the 28 Bonferroni-significant neighborhoods? Do many of these neighborhoods overlap in some capacity, and/or do these 28 neighborhoods capture distinct subsets of variants? Can neighborhood signals be clustered in order to highlight associations coming from distinct gene sets/functional pathways?*

Response: We thank the reviewer for the questions. On average, each pair of the top 28 neighborhoods share 705 out of 1000 variants. We have added this information to the “*ENSAS captures proband-sibling local GC content differences in specific gene-expression-based neighborhoods*” results section:

We tested a total of 29,820 expression neighborhoods and after a Bonferroni-based multiple testing correction we observed 28 neighborhoods with a significant difference ($p \leq 1.7 \times 10^{-6}$) in local GC content between proband and siblings. **On average, each pair of the 28 neighborhoods show considerable overlap, sharing 705 out of 1000 variants.**

In addition, to examine the distinction between these 28 neighborhoods we clustered them using DBSCAN², a suitable clustering technique for identifying outliers, based on Manhattan distance between neighborhoods' variants. We obtained five clusters, with one containing 24 neighborhoods and the other four containing exactly one neighborhood each. These four neighborhoods have on average 529, 542, 587 and 462 variants overlapping with neighborhoods in the largest cluster, and average Jaccard index of 0.35, 0.35, 0.40, 0.30 with neighborhoods in the largest cluster, respectively. We observed consistency for the top enriched terms across the clusters, with terms broadly related to synaptic transmission and cell junction being among the top 5 enriched terms for their respective categories in each cluster. We added this analysis as the section “*Clustering of the top neighborhoods identified by ENSAS*” in the Supplementary Text:

To examine the extent by which the top 28 neighborhoods characterize diverse genes and pathways, we clustered these neighborhoods using DBSCAN² based on their variant compositions (Methods) and on each cluster repeated the GO enrichment analysis on the union of genes of all neighborhoods in the cluster, using the two different backgrounds described above. We observed that one cluster contained 24 neighborhoods while the remaining four clusters contained one neighborhood each. These four neighborhoods had on average between 462 and 587 variants overlapping with neighborhoods in the largest cluster, and on average a Jaccard index between 0.30 and 0.40 with neighborhoods in the largest cluster. The clusters showed consistency for the top enriched terms, with terms broadly related to synaptic transmission and cell junction being significant in all clusters (Supplementary Figure 3). This suggests that the top neighborhoods identified by ENSAS implicate similar biological processes.

The above results are referred to by the following new section added to the “*ENSAS captures proband-sibling local GC content differences in specific gene-expression-based neighborhoods*” results section in the main manuscript:

In addition, we examined the extent to which the top 28 neighborhoods characterized diverse genes and pathways. To do this we first clustered these neighborhoods using DBSCAN² based on their variants (Methods) and then for each cluster repeated the GO enrichment analysis on the union of genes of all neighborhoods in the cluster. We observed similar terms enriched across the clusters, suggesting that the top neighborhoods identified by ENSAS implicate similar biological processes (Supplementary Text, Supplementary Figure 3).

We updated Supplementary Figure 3 to include a plot for this analysis:

Supplementary Figure 3 Top 5 enriched GO terms under each GO category for clusters of the top 28 significant neighborhoods in the M-F upstream variants ENSAS analysis, using the union of all genes assigned to the M-F upstream variants as background (left) or the union of genes assigned to the M-F upstream variants that are also differentially expressed within any of the 13

brain-related GTEx tissues as background (right). x-axis shows Benjamini-Hochberg adjusted Fisher's exact p-values. Vertical dashed line shows p-value significance threshold of 0.05. **(a)** Median p-values across neighborhoods in the largest cluster with 24 neighborhoods. Error bars show interquartile range across the neighborhoods. **(b-e)** P-values for the remaining clusters, each containing one neighborhood.

We also added a methods section named “Clustering of the top neighborhoods using DBSCAN”:

Clustering of the top neighborhoods using DBSCAN

We clustered the top 28 neighborhoods using DBSCAN², with the distance metric being the Manhattan distance between the variants of each pair of neighborhoods, which is equal to the neighborhood size M (1000) minus the number of variants shared by the two neighborhoods. Each cluster is allowed to have as few as one neighborhood, and neighborhoods assigned to the same cluster must have a pairwise distance less than 500, corresponding to half of the neighborhood size, to all other neighborhoods in the same cluster.

(5) *The test for signal differences between sibling pairs sequenced in the same vs. different lanes is an appreciated QC check, though I am surprised by the lack of signal from the lane-mismatched pairs (if sequencing lane is not a factor, one might expect similar performance from both sets). What is the median p-value for the top 28 neighborhoods in the lane-mismatched sets? Are there any known (metadata) factors associated with whether probands*

Response: We thank the reviewer for the questions. Despite having a weaker signal compared to the lane-matching samples, the lane-mismatched samples had a median p-value of 0.0047 for the 28 neighborhoods significant for the full set, and 7 out of 28 of them were among the top 30 most significant neighborhoods within the lane-mismatched samples (Supplementary Figure 4c). We added this information to the “*ENSAS Proband-sibling local GC content differences not driven by obvious sequencing batch effects or promoter variants*” results section:

No significant neighborhoods were identified when restricted to the samples with mismatched sequencing lanes (Supplementary Figure 4c) and the overall distribution of the p-values appeared relatively uniform (Supplementary Figure 4a). However, the neighborhoods significant in the full set of samples had a median p-value of 0.0047 in this subset of samples and 7 out of 28 of them were among the top 30 most significant neighborhoods (Supplementary Figure 4c). Notably the neighborhood p-values from both the lane-matching and lane-mismatched subsets showed high correlation with those from the full set of samples (Spearman's $r=0.78$ and 0.72 respectively, Supplementary Figure 4b,c). Overall, while the signal can be mainly attributed to the subset of lane-matching samples whose comparison would be expected to be less susceptible to technical sequencing confounders, the association from the lane-mismatched samples is in a consistent direction.

Whether the sample pairs have matching or mismatched sequence lanes does associate with the sequencing phases. In total 900 out of the 968 samples from pairs with matching sequencing lanes were sequenced in phases 2 and 3-1 while 590 out of the 606 samples from pairs with mismatched sequencing lanes were sequenced in phases 1, 3-2, and the pilot phase. We have now included this information in Supplementary Table 3:

Supplementary Table 3 Number of samples from male proband-female sibling pairs, with matching or mismatched sequencing lane information, in each of the sequencing phases

	Pilot	Phase1	Phase2	Phase3-1	Phase3-2
Matching lanes	4	40	516	384	24
Mismatched lanes	32	404	10	6	154

In the revised manuscript we repeated the ENSAS analyses for each phase or subphase excluding the pilot, and observed the signal concentrated largely in phases 2 and 3, with phase 2 having a more significant presence of the signal. We added descriptions of these analyses to the section “*ENSAS Proband-sibling local GC content differences stratified by sequencing phases*” in the Supplementary Text:

We note that whether the samples have matching or mismatched sequencing lanes is associated with their sequencing phases. Overall, the samples were sequenced across three primary phases, with phase 3 having two sub-phases. In total, 900 out of the 968 samples from pairs with matching sequencing lanes were sequenced in phases 2 and 3-1, while 590 out of the 606 samples from pairs with mismatched sequencing lanes were sequenced in phases 1, 3-2, and the pilot phase (Supplementary Table 3). When we repeated the tests on each of these phases excluding the pilot phase (Supplementary Figure 4d-i), we observed that the overall signal was mainly concentrated in phase 2 (526 samples, Supplementary Figure 4d) and to a lesser extent phase 3 (568 samples, when combining both 3-1 and 3-2 subphases, Supplementary Figure 4d). The 28 brain-related neighborhoods significant in the full set of samples have median p-values of 8.2×10^{-5} and 4.6×10^{-4} in phases 2 and 3 respectively (Supplementary Figure 4f, i).

Supplementary Figure 4 Comparison of local GC content Mann-Whitney U p-values for ENSAS on selected subsets of M-F upstream variants of all samples. Horizontal and vertical dashed lines show Bonferroni-based p-value multiple testing significance thresholds of $0.05 / n$ where $n=29,820$ is the number of neighborhoods. Diagonal dashed lines show the unit slope. **(a-c)** Samples with matching sequencing lanes with their probands/siblings (“matching lanes”), and samples with mismatched sequencing lanes with their probands/siblings (“mismatched lanes”) are shown **(a)** QQ-plots for the neighborhood local GC content p-values of three sample groups. **(b)** Neighborhood p-values for all samples vs. samples with matching sequencing lanes.

(c) Neighborhood p-values for all samples vs. samples with mismatched sequencing lanes. (d-i) Samples from sequencing phases 1, 2, 3 (with two subphases, 3-1 and 3-2) are shown. (d) QQ-plots for the neighborhood local GC content p-values of each of the phases. (e-i) Neighborhood p-values for all samples vs. samples in each of the phases.

To summarize, the signal can be mainly attributed to samples with matching sequencing lanes, but samples with mismatched sequencing lanes also showed some extent of a similar signal. Moreover, the signal is distributed across two sequencing phases, with a stronger presence in phase 2. Our overall conclusion is that the signal is not explained by any obvious sequencing technical factors and we added this remark following the above paragraph:

Overall, these analyses support there is no obvious explanation in terms of technical confounding. First, while the signal can be mainly attributed to the subset of lane-matching samples, those comparisons would be expected to be less susceptible to technical sequencing confounders and the association from the additional samples is in a consistent direction. Additionally, while phase 2 had the largest number of lane-matched samples and was most associated, the overall association was enhanced by considering samples from additional phases.

All these results are referred to by the following new section added to the “*ENSAS Proband-sibling local GC content differences not driven by obvious sequencing batch effects or promoter variants*” results section in the main manuscript:

We noticed that whether proband and sibling samples have matching or mismatched sequencing lanes is associated with their sequencing phases (Supplementary Text, Supplementary Table 3). We thus repeated the tests on each of these phases excluding the pilot phase, which had limited samples. We observed the most significant associations from phase 2, which also had the largest number of lane-matched samples. However, the overall association was enhanced by considering samples from additional phases (Supplementary Text, Supplementary Figure 4d-i).

(6) *The ENSAS analysis run excluding putative promoter variants is interesting, and demonstrates ENSAS' ability to discover new associations. But I am also curious, how does ENSAS perform for just the putative promoter variants? Does the ENSAS approach also identify an association signal for these variants? (validation of An et al?)*

Response: We thank the reviewer for the questions. We applied ENSAS to 898 M-F upstream promoter variants, treating them as if they were in a single neighborhood. We did not observe the signal, as the resulting local GC content p-value is 0.17.

We note there are at least two reasons the promoter signal reported by An *et al.* does not necessarily replicate in our ENSAS analysis. First, the promoter signal by An *et al.* was based on the proband-sibling difference in total variant counts. The ENSAS framework focuses on a

different aspect, specifically testing differences in the distribution of sequence properties between proband and sibling variants. In addition, while both An *et al.* and Zhou *et al.* used overlapping samples there are differences in the exact set of samples each used and further the *de novo* variant calling and filtering pipelines are different (for example, only Zhou *et al.* removed all variants overlapping repeats), thus the resulting sets of *de novo* variants are not the same.

We edited the second paragraph in the “*ENSAS Proband-sibling local GC content differences not driven by obvious sequencing batch effects or promoter variants*” section to make a remark on the previous promoter signal:

That we were able to observe the signal after excluding the promoter regions is expected as most variants we considered are outside of promoter regions. We also note that our analysis is not directly comparable to the previously reported promoter association analysis, since the previous promoter association analysis was based on a different analytical framework that did not directly test for sequence differences between proband and sibling variants and used a different set of *de novo* variant calls.

(7) *What is the rationale for testing up to 7 bp k-mers? Is a 7 bp k-mer long enough to capture relevant sequence context, e.g. functional elements like transcription factor binding sites? Also, are these specific k-mers sequences that are most strongly associated with the association signals?*

Response: We thank the reviewer for the questions. Using longer k-mers poses both computational and statistical difficulties. Computationally, longer k-mers come with an exponential increase in both memory and runtime. Statistically, with fewer observations of each k-mer it becomes difficult to effectively fit a model. Consistent with this we observed that when we increased the k-mer length from 6 to 7 we already observed a decrease in overall predictive performance, as shown in Figure 4. There exist alternative representations of sequence such as gapped k-mers or deep learning representations that in principle could capture TFBS that are longer, but such models would be more difficult to fit with the size of our training data and the level of signal within it. Our focus with ENSAS is to evaluate how predictive relatively simple models of sequence are.

To examine the predictive power for each of the 6-mers, for each of the top neighborhoods we tested if each 6-mer is enriched for proband variants or sibling variants using binomial tests, described in the “*ENSAS: Sequence analysis of the gene expression neighborhoods*” methods section:

To investigate the significance of individual k-mers for each of the top 28 neighborhoods with significant proband-sibling local GC content differences, within each neighborhood for each 6-mer we used a two-sided binomial test to test if the 6-mer is significantly more prevalent in the proband variants or the sibling variants. We controlled for multiple testing using a Bonferroni-corrected p-value threshold of $0.05/4^6$.

After correcting for multiple testing, we did not observe any 6-mer significantly enriched in probands or siblings across the top neighborhoods (Supplementary Figure 7a). The directionalities of the 6-mer enrichments reflect the proband-sibling GC content differences, as high-GC 6-mers are more enriched in probands while low-GC 6-mers are more enriched in siblings (Supplementary Figure 7b). 6-mers containing only As or Ts, on average, have more significant p-values than other 6-mers (Supplementary Figure 7c,d). We added a brief remark for this analysis in the “*Investigating sequence signal beyond local GC content in the significant neighborhoods*”:

When the random train-test splitting was repeated 100 times for each of these 28 neighborhoods, the 6-mer Naive Bayes model had better median performance in 18 of these neighborhoods (Supplementary Figure 7a). We also directly tested whether individual 6-mers exhibited significant difference in prevalence between proband and sibling variants within these neighborhoods (Supplementary Figure 7b). We did not observe any individual 6-mer reaching statistical significance. However, as would be expected by the results with GC content, we did observe that high-GC 6-mers are more enriched in probands while low-GC 6-mers are more enriched in siblings (Supplementary Figure 7c). We further observed that 6-mers containing only A's and T's have more significant p-values on average than sets of 6-mers with other frequencies of A's and T's (Supplementary Figure 7d).

We updated Supplementary Figure 7 to include plots for these analyses (Supplementary Figure 7b-d):

Supplementary Figure 7 K-mer-based analysis for each of the 28 significant neighborhoods.

(a) 6-mer Naive Bayes model (NB) and local GC content (GC) Mann-Whitney U p-values across 100 random train-test splits for each of the 28 significant neighborhoods. Boxes show median and interquartile ranges. Points above or below 1.5x interquartile range are drawn as outliers. The neighborhoods are sorted by their median Naive Bayes p-values. **(b)** Proband-sibling 6-mer enrichments across the 28 significant neighborhoods. Each point represents a unique 6-mer, colored based on its GC content. The x-axis is $\log_{10}((6\text{-mer count in proband variants} / 6\text{-mer count in sibling variants}) / (\text{number of proband variants} / \text{number of sibling variants}))$, with the mean taken over 28 neighborhoods and larger values indicating preference for proband variants. The y-axis is the two-sided binomial p-value of the 6-mer's occurrence count in proband variants vs. sibling variants with the median taken over 28 neighborhoods. Horizontal dashed line shows Bonferroni-based p-value multiple testing significance threshold of $0.05 / n$ where $n=4^6$ is the total number of 6-mers. **(c)** Distributions of the mean log ratios (same as the x-axis in (b)) for 6-mers with different GC contents. **(d)** Distributions of the median binomial p-values (same as the y-axis in (b), but displayed on a different scale) for 6-mers with different

GC contents. For both (c) and (d) boxes show median and interquartile ranges. Points above or below 1.5x interquartile range are drawn as outliers.

Overall we did not find individual 6-mers significantly associated with the signal. A possible explanation is that the GC content-based signal is distributed across many 6-mers while each individual 6-mer only has a small contribution. We also did not find significant evidence of 6-mer or other k-mer models outperforming local GC content in terms of proband-sibling association signal. However, we cannot exclude the possibility that alternative predictors based on k-mer features or alternative predictors based on other sequence features would do so.

We added a remark in the same section:

Despite the results we observed in the k-mer Naive Bayes model and the individual 6-mer analysis, we cannot exclude the possibility that there exists additional sequence association signals beyond local GC content that could be detected with other predictive models based on k-mer features or other sequence representations.

(8) Page 23, lines 13-20 proposes a hypothesis that chromatin or transcriptomic differences between males and females may account for the increased signal in male proband-female sibling pairs. Can the authors clarify how such a mechanism could work? Is the implication that variants in the same neighborhoods in males and females may have different expression profiles, and therefore different effect sizes in males and females?

Response: We thank the reviewer for the questions. We proposed the hypothesis as several prior studies showcase the potential role of sex differences in transcription or chromatin environment during the development that could be relevant to sex differences in ASD. For example, previous research integrating samples from SSC and other ASD studies indicated that genes containing putatively damaging coding *de novo* mutations show higher expression in prenatal female brain regions than in males, suggesting potential compensatory effects in females³. Another study reported elevated levels of the repressive chromatin mark H3K27me3 in female placenta compared to their male counterparts, which reduces female fetuses' vulnerability to gene expression disruptions in the developing hypothalamus, a region associated with ASD⁴. We added these examples to the discussion:

Potentially related, previous research found genes with putatively damaging *de novo* coding variants are more highly expressed in prenatal female brains than in males, suggesting potential compensatory effects in females³. Also potentially relevant, another study reported elevated levels of the repressive chromatin mark H3K27me3 in female placenta compared to males, which reduces female fetuses' vulnerability to gene expression disruptions in the developing hypothalamus, a region associated with ASD⁴. Another avenue for future investigation is to understand why the observed signal is specific to variants in regions upstream of TSS relative to downstream and the extent to which it might be related to different transcriptional processes or chromatin environments associated with these regions.

While it is possible as the reviewer suggests that *de novo* mutations in the same neighborhood are associated with different expression profiles in males and females, this is not the only possible explanation. In particular, it is possible the specific observed male-female difference in our analyses may not be directly causal to the ASD but instead could be associated with an additional unobserved causal mechanism, which may also contribute to the previously observed chromatin and transcriptomic differences. We added the following remark to the discussion:

We also note that this association signal that we identified does not necessarily imply a causal relationship with ASD. It is possible for instance that this signal along with the previously established sex differences in transcription or prenatal chromatin environment are all tied to a common underlying causal mechanism.

Minor comments

(1) Fig 1a) shows that DNA DIS increases as GC content increases, though the x-axis "GC content bins" is not intuitive to interpret. Can this be converted to % GC content, or perhaps a second x-axis added to show % GC content? (same for Supp Fig 1a)

Response: We thank the reviewer for the comment. Local GC content is calculated based on the 201-bp window centering each variant, therefore it can only take values between 0 to 201. In Figure 1a and supplementary Figure 1a, we stratified the variants based on their local GC content into intervals of 10 and each "bin" represents an interval, from 0-10 to 190-201. For Figure 1 we updated the x-axis to show local GC content percentages and added clarifications in the figure captions:

Figure 1 DIS vs. local GC content. **(a)** The y-axis corresponds to the DNA DIS score and the x-axis evenly sized local GC content intervals, each with size 10 (corresponding to 5%, with bin labeled as N representing N%-5% - N%) . Each interval is represented by a box showing the distribution of its variant's DIS. The boxes correspond to the quartiles, the lengths of whiskers correspond to 1.5x interquartile range and variants above/below the whiskers are defined as outliers. Spearman correlations are computed between DIS and local GC content, across all variants. **(b, c, d)** Proband-sibling differences for DIS (combined) or local GC content of variants assigned to each of the 53 GTEx tissue or cell types are shown. Both the x- and y-axis show $-\log_{10}$ p-values from one-sided Mann-Whitney U-tests. Horizontal and vertical dashed lines show p-values at FDR threshold of 0.05. Points greater than (but not on) these lines are significant after FDR correction. Diagonal dashed lines show the unit slope. **(b)** Proband-sibling differences in local GC content vs. proband-sibling differences for DIS (as conducted by Zhou et al.) **(c)** Male proband-female sibling pair differences for local GC content vs. male proband-male sibling pair differences for local GC content, with coding and CSS variants removed; **(d)** Male proband-female sibling pair differences for local GC content in variants <100kbp upstream of nearest outermost TSS only vs. in variants <100kbp downstream of nearest outermost TSS, with coding and CSS variants removed. Additional related panels can be found in Supplementary Fig 1.

We also updated Supplementary Figure 1 accordingly (shown in our response to comment #1).

Fig 1b) x-axis is labeled as "DIS -log10(p)" but figure legend notes that x-axis is the proband - sibling difference in DIS. Please re-label the x-axis to clarify this.

Response: We thank the reviewer for the comment. "DIS -log10(p)" represents the -log10 p-values from one-sided Mann-Whitney U-tests on proband vs. sibling DIS. This is clarified in the figure legends: "Proband-sibling differences for DIS (combined) or local GC content of variants assigned to each of the 53 GTEx tissue or cell types are shown. Both the x- and y-axis show -log10 p-values from one-sided Mann-Whitney U-tests." We updated the x- and y-axis in Figure 1b and Supplementary Figures 1b, c to say "Proband-sibling" differences.

Fig 1b) typo in legend text, repeated text "proband sibling differences"

Response: We thank the reviewer for pointing this out. This has been corrected.

(2) The finding that local GC content can identify noncoding association signals is interesting, and the authors note in Results that "these analyses also do not explain why local GC content, which can be both biologically significant and a confounder in genomic analyses, was sufficient to identify the association signals". If space allows, can the authors expand on these possible mechanisms, biological and technical, in the Discussion?

Response: We thank the reviewer for the suggestion. We have added some examples in the discussion section. Specifically, we mention the previously established correlation between GC content and read coverage in sequencing studies⁵, and the association between GC content and biological features such as promoters⁶, enhancers⁷, gene structure⁸ and mRNA decay⁹.

A caveat in interpreting the results of ENSAS is that while GC content can be associated with biological features such as promoters⁶, enhancers⁷, gene structure⁸, and mRNA decay⁹ it can also correlate with sequencing technical factors. For example, sequencing read fragments can be biased towards GC-rich regions, leading to positive correlation between GC content and sequencing coverage⁵. As a result ENSAS does not directly determine if sequence differences are due to biological differences or technical confounders associated with sequencing.

(3) Page 14, paragraph ~lines 14-35 describes GO enrichment analysis for the 28 neighborhoods, but the approach is confusing as described. Can the authors clarify, were the genes in each neighborhood tested separately for GO enrichment? Or was the union of genes across all neighborhoods tested for enrichment?

Response: We thank the reviewer for the questions. In the GO enrichment analysis, each neighborhood was tested separately. We reworded a sentence in the "ENSAS captures proband-sibling local GC content differences in specific gene-expression-based neighborhoods" results section to clarify this:

To biologically characterize the sets of genes associated with the top neighborhoods in the previous M-F upstream variants analyses we conducted a Gene Ontology (GO) enrichment analysis for genes associated with each of the 28 neighborhoods (Figure 3a, Methods). For this analysis we used as background the set of all genes assigned to M-F upstream variants, **and separately for each neighborhood tested if the genes in the neighborhood showed enrichment in GO terms compared to the background gene set (Methods).**

We also added a remark in the “*Gene ontology enrichment analysis for the top neighborhoods*” methods section to emphasize that each neighborhood was tested separately:

Gene ontology enrichment analysis for the top neighborhoods

We performed Gene Ontology (GO) enrichment analyses for the top neighborhoods with the most significant local GC content p-values from the ENSAS performed on M-F upstream variants. For the GO analyses, **we tested each neighborhood separately.** We used as the foreground the set of genes with assigned variants in each neighborhood and two different sets of background genes: (1) union of all genes assigned to the M-F upstream variants and (2) the union of genes assigned to the M-F upstream variants that were also considered differentially expressed within any of the 13 brain-related GTEx tissues

(4) Page 20, line 58 “we observed substantially reduced p-values” - should this read “increased” p-values (less significant)?

Response: We thank the reviewer for pointing this out. This sentence has been changed as shown below:

We observed substantially **less significant** p-values in these analyses compared to the M-F upstream analysis (Supplementary Figure 9).

Reviewer #2: The topic of the effect of non-coding de novo mutations in ASD has bit hotly debated fueled by the preponderance of non-coding variants among common variants influencing heritability and by the unambiguous effect of coding de novo mutations. The community has not settled on whether non-coding de novo mutations are also a significant contributor to the ASD risk (although the question may be soon resolved with the widely anticipated new sequencing data). This manuscript has three components.

First, the authors demonstrate that previously reported apparently positive findings facilitated by a ML method can be replicated (and even enhanced) by substituting the ML machinery by a measure of GC content. Although this result is not glorious, I find it important. It reminds us that in some cases a ML black box lends itself to a simple interpretation.

Second, it is observed that the effect is limited to male-female proband-sibling pairs and to the mutations upstream of genes. These are highly surprising observations (especially, the former). It goes against the naïve expectation.

Third, the authors developed the ENSAS software. Even though it is an interesting development (and perhaps is where the larger part of the work went), it is probably the least impactful aspect of the paper. Application of ENSAS have not led to new observations, and it is not very likely that the method will be widely adopted by the field.

My specific comments are listed below.

Response: We thank the reviewer for the summary and positive comments. We also thank the reviewer for the constructive comments, which has led to a strengthened manuscript, and to which we respond below.

1. The manuscript reports many p-values corresponding to numerous statistical analyses. The major critique of the field stems from the multitude of tests including definitions of regulatory regions, loci etc. It would be great seeing one flagship number, e.g. p-value corrected for all tests performed in this work.

Response: We thank the reviewer for the comment. We agree that the handling of multiple tests is an ongoing issue in the field, especially in scenarios where a multitude of genomic regions are tested. However, we note that our analyses involve distinct sets of statistical tests. In addition as discussed in the next comment, it inherits analysis choices from prior work. Applying a uniform multiple testing correction across all these tests may oversimplify their differing statistical frameworks and test samples and may not fully account for all effective tests done.

Instead, we currently apply multiple testing corrections independently for each family of tests, which is commonly done in the field, and now acknowledge this point in the discussion section:

We also note caveats in our analysis related to statistical considerations. A major issue in whole-genome sequencing studies is the handling of multiple tests. In our study, we employed

distinct sets of tests that differed in their frameworks and test samples. We applied multiple testing corrections independently for each set instead of a uniform correction across all sets to preserve these differences, but our reported p-values do not reflect multiple testing across different families of analyses.

2. I do not doubt the statistical discipline in this work. However, it inherits a lot of ad hoc parameter choices from previous studies that may have (unintentionally) lead to the inflation of the signal (at the end only positive findings out of many attempts are published). Some discussion of this would be warranted, especially given that we anticipate a new data release potentially resolving the whole issue.

Response: We thank the reviewer for raising the point. It is the case that we followed test procedures and parameter settings of Zhou *et al.* where applicable for consistency. However, we emphasize that one of the main findings of this manuscript, that the brain expression signal is concentrated in a specific subset of variants (from male proband-female sibling pairs, upstream of TSS) was not reported by Zhou *et al.* and there is no reason to believe directly influenced their parameter choices. We feel that readers can have greater confidence in results here compared to if they were published in the original paper since parameter choices were published by one group before being used by a different group to obtain key results unreported by the first group. However, we agree that we currently cannot exclude the possibility that the associations reported here may fail to replicate in larger datasets. We summarize these points by adding the following sentence to the discussion section:

We also recognize that our analysis inherited test procedures and parameters that already generated positive findings in the previous study by Zhou *et al.*¹, and therefore our observed signal can be subject to inflation. Although a main finding that the brain expression signal is concentrated in a specific subset of variants (male proband-female sibling pairs, upstream of TSS) was not reported by Zhou *et al.*¹, we still cannot exclude the possibility that the signal is driven by technical factors and requires replication in larger samples to increase confidence in its biological significance.

3. I do not fully understand the choice of the test (Mann-Whitney on GC content). Is the expectation that a single large effect *de novo* mutation contributes to a subset of cases? The manuscript would benefit from a simulation (or an analysis) to show that the observed signal is in basic agreement with the known epidemiology of ASD and that the selected test is appropriate.

Response: We thank the reviewer for the comment. Our choice of test was driven by the previous study by Zhou *et al.* which applied Mann-Whitney U-test to test for proband-sibling DIS differences, and we thus used the Mann-Whitney U-test for consistency, as mentioned in our response to comment #2. The choice of the test was not driven by the specific expectation for single large effect *de novo* variants. Since the disease mechanisms of *de novo* noncoding variants in ASD remain largely unexplored it would be more reasonable to employ a flexible test, instead of a test with specific biological assumptions. Under these criteria, we argue that the choice of the test is appropriate, as the Mann-Whitney U-test does not make strong assumptions about the underlying distribution of the data, as opposed to parametric tests such as the t-test. We also note that it is possible that the

signal we may be detecting might not be the actual causal mechanism but a signal associated with it in which case a wider range of observed signals might be possible. We discussed this in the following paragraph added to the discussion section:

We also note that this association signal that we identified does not necessarily imply a causal relationship with ASD. It is possible for instance that this signal along with the previously established sex differences in transcription or prenatal chromatin environment are all tied to a common underlying causal mechanism.

In the manuscript, we added a remark in the “*Local GC content largely explains noncoding ASD associations attributed to deep learning*” results section to mention that the Mann-Whitney U test is the same test employed by Zhou et al.:

Using local GC content we obtained an association signal with an overall similar distribution of p-values for the DNA DIS across the 60 curated variant sets (two-sided Mann-Whitney U $p=0.16$, the same test as in Zhou et al.¹) and the combined DIS across the 53 GTEx tissues ($p=0.98$).

4. The observations that the signal is limited to the sequence upstream of TSS (but not promoter!) and to the M-F comparisons are highly intriguing (assuming they are not statistical artifacts). The difference between the M-M and M-F comparisons needs more discussions. I see two possible directions of inquiry (there are probably more that I do not see). One is that some of the mutations are post-zygotic, and there is a bias in somatic mutagenesis between males and females. This is discussed in the manuscript. This explanation is inconsistent with the analysis of the deCODE data. The other possibility is that due to some protective effect in females, brothers of probands are more likely to be affected than sisters. This would mean that the polygenic risk in probands with an unaffected sister is, on average, larger than in probands with an unaffected brother (the effect is probably small and may lead to the opposite direction of the effect but may still be explored).

Response: We thank the reviewer for the comment. We recognize the possibilities for both explanations suggested by the reviewer. For the first explanation, while we did not observe a baseline male-female sequence difference within the deCODE Icelandic dataset, we saw nominally significant difference in local GC content for male sibling vs. female sibling variants near brain expressed genes, as discussed in our response to reviewer #1 comment 2. This has been added to the “*Local GC content differences between male and female samples*” section in the Supplementary Text:

To assess the extent to which the observed signal might be associated with general differences between male and female samples, we repeated the analysis by comparing male siblings with female siblings, as well as comparing male probands with female probands. For both comparisons, no tissue reached the FDR-based significance threshold of 0.05. Between variants from male siblings ($n=15,685$) and female siblings ($n=17,272$) we note that 10 out of the 13 brain-related tissues are nominally significant ($p < 0.05$), (Supplementary Figure 1f). Between variants from male probands ($n=30,129$) and female probands ($n=4,496$) one brain tissue was nominally significant ($p < 0.05$,

Supplementary Figure 1f) though we note the sample size of variants for female probands is limited in this analysis. These results provide suggestive evidence of differences between male siblings and female siblings though it did not reach the same level shown by the male proband-female sibling analysis.

We also note that the SSC dataset is different as it contains samples that are already conditioned on being in a family with ASD. The observed sequence difference may only manifest when ASD is in the family, instead of for any random sample drawn from the population. Following the above text we added discussions for this:

We emphasize that the SSC dataset only consists of samples from ASD families and thus siblings are not necessarily representative of unaffected samples in the general population.

For the second explanation, we recognize that previous studies have shown signs of the female protective effect. As mentioned in our response to reviewer 1 comment #8, these protective mechanisms can operate at the expression³ or chromatin level⁴. Following up on the remark on the relationship between our observations and polygenic risk scores would be an interesting direction for future research, but we consider it outside the scope of this study which would require access to common variant data for that analysis purpose. We also note that the observed male-female signal may not be directly causal to the ASD phenotype but stems from the same biological basis as these previously discovered male-female chromatin or expression differences. We acknowledge this in the discussion:

We also note that this association signal that we identified does not necessarily imply a causal relationship with ASD. It is possible for instance that this signal along with the previously established sex differences in transcription or prenatal chromatin environment are all tied to a common underlying causal mechanism.

5. Unless I missed it in the manuscript, is there any difference between M-M and M-F pairs in the prevalence of coding *de novo*s?

Response: We thank the reviewer for the question. We are uncertain if the reviewer is asking about the difference between number of variants in probands and siblings, but within M-M and M-F pairs, or the difference between total number of variants M-M and M-F pairs regardless of proband-sibling status.

We first compared the number of *de novo* coding variants between probands and siblings, and then performed the same analysis for protein-truncating variants (PTVs) which showed association signal in An *et al.*¹⁰ When comparing all proband to all sibling samples in the dataset, we did not observe significant difference in the number of *de novo* coding variants

(two-sided binomial p-value=0.25). Also no significant differences were found when restricted to M-F pairs (two-sided binomial p-value=0.20) or M-M pairs (two-sided binomial p-value=0.67).

We added this information to the results section:

Given the male proband-female sibling differences for non-coding variants, we were interested to know if similar male proband-female sibling differences can be found with respect to the number of *de novo* coding variants or protein-truncating variants (PTVs) (Methods), the latter of which previously showed ASD association signal in the previous study by An *et al.*¹⁰, For coding variants we did not observe any significant proband-sibling difference (Supplementary Table 1).

For PTVs, following Werling *et al.*¹¹ and An *et al.*¹⁰ we adjusted the *de novo* variant counts based on paternal age. We observed that the probands show significantly higher occurrence compared to siblings (two-sided binomial p-value= 6.3×10^{-4} , fold enrichment=1.20) as previously reported by An *et al.*¹⁰ and the difference remained significant after adjusting for paternal age (two-sided binomial p-value= 5.7×10^{-4} , fold enrichment=1.21, Methods). The signal is more significant when considering all samples compared to when restricted to M-F pairs (two-sided binomial p-value=0.12, fold enrichment=1.14 before adjusting for paternal age, p-value=0.12, fold enrichment=1.14 after adjustment) or M-M pairs (two-sided binomial p-value= 4.8×10^{-3} , fold enrichment=1.27 before adjusting for paternal age, p-value= 7.2×10^{-3} , fold enrichment=1.27 after adjustment). We added this information to the results section:

For PTVs, we did observe significant proband-sibling differences ($p < 0.05$) when considering all probands and all siblings and all male proband-male sibling pairs, while male proband-female sibling pairs showed an enrichment that did not reach statistical significance (Supplementary Table 1). These results suggest the association found for non-coding variants among male proband-female sibling pairs is not directly reflected in the PTV count association signal.

We described the PTV definition in the new Methods section shown below:

Defining PTVs

We annotated the *de novo* variants using the Ensembl Variant Effect Predictor for GRCh37. We defined PTVs to be those annotated as “stop_gained”, “splice_donor_variant”, “splice_acceptor_variant” or “frameshift_variant”, following the definitions previously used by Fu *et al.*¹²

The testing procedure is added to the Methods section:

Enrichment tests for *de novo* variant and PTV counts

For each sample we counted the number of *de novo* variants and PTVs. Following Werling *et al.*¹¹ and An *et al.*¹⁰ we adjusted the *de novo* variant counts based on paternal age. We performed a linear regression using the paternal age of each sample as the predictor variable and the total number of *de novo* variants of each sample as the response variable. The residual of the regression model was then shifted such that its

sample-wise mean is the same as the mean number of *de novo* variants across samples before adjustment. This shifted residual was taken as the new adjusted counts. For each sample, we computed the ratio between adjusted and raw counts. To get the adjusted counts of *de novo* variants and PTVs, their respective raw counts were multiplied by this sample-specific ratio. We used two-sided binomial tests to test for proband-sibling differences in *de novo* counts.

For the alternative interpretation, we also tested for differences between M-M and M-F pairs in the total number of *de novo* coding variants and *de novo* PTVs. We did not observe a significant difference in the number of *de novo* coding variants between M-F and M-M pairs (two-sided binomial p-value=0.44). Neither did we observe significant difference in the number of protein-truncating variants (two-sided binomial p-value=0.84).

References

1. Zhou, J. *et al.* Whole-genome deep-learning analysis identifies contribution of noncoding mutations to autism risk. *Nat. Genet.* **51**, 973–980 (2019).
2. Ester, M., Kriegel, H.-P. & Xu, X. A Density-Based Algorithm for Discovering Clusters in Large Spatial Databases with Noise. *KDD-96 Second Int. Conf. Knowl. Discov. Data Min.* (1996).
3. Zhang, Y. *et al.* Genetic evidence of gender difference in autism spectrum disorder supports the female-protective effect. *Transl. Psychiatry* **10**, 4 (2020).
4. Nugent, B. M., O'Donnell, C. M., Epperson, C. N. & Bale, T. L. Placental H3K27me3 establishes female resilience to prenatal insults. *Nat. Commun.* **9**, 2555 (2018).
5. Benjamini, Y. & Speed, T. P. Summarizing and correcting the GC content bias in high-throughput sequencing. *Nucleic Acids Res.* **40**, e72–e72 (2012).
6. Fenouil, R. *et al.* CpG islands and GC content dictate nucleosome depletion in a transcription-independent manner at mammalian promoters. *Genome Res.* **22**, 2399–2408 (2012).
7. Azoifeifa, J. G. *et al.* Enhancer RNA profiling predicts transcription factor activity. *Genome Res.* **28**, 334–344 (2018).

8. Amit, M. *et al.* Differential GC Content between Exons and Introns Establishes Distinct Strategies of Splice-Site Recognition. *Cell Rep.* **1**, 543–556 (2012).
9. Courel, M. *et al.* GC content shapes mRNA storage and decay in human cells. *eLife* **8**, e49708 (2019).
10. An, J.-Y. *et al.* Genome-wide de novo risk score implicates promoter variation in autism spectrum disorder. *Science* **362**, eaat6576 (2018).
11. Werling, D. M. *et al.* An analytical framework for whole-genome sequence association studies and its implications for autism spectrum disorder. *Nat. Genet.* **50**, 727–736 (2018).
12. Fu, J. M. *et al.* Rare coding variation provides insight into the genetic architecture and phenotypic context of autism. *Nat. Genet.* **54**, 1320–1331 (2022).

Second round of review

Reviewer 1

I appreciate the authors' thorough and thoughtful consideration of reviewers' concerns. I have a few remaining questions or notes:

- (1) Fig 1a, is the Y axis scale correct? Looks like DNA DIS scores ceiling out around a value of 4?
- (2) Bottom of page 8, thank you for adding the number of families alongside the number of variants being tested for each proband-sibling combination. Can the authors adopt this consistently throughout the text, e.g. mid-page 9, newly added text noting results from female pro-female sib and female pro-male sib pairs? (and there may be others)
- (3) Supp Fig 7a - X axis labels are gene names, can the authors clarify in the legend that this is how neighborhoods are named, e.g. "28 significant neighborhoods labeled by their central gene"
- (4) Thank you for adding the note about the number of variants shared between neighborhoods. I assume then that there is also considerable sharing of genes between neighborhoods? I apologize for not considering this in depth on the first submission, but what is the rationale for not merging these highly overlapping neighborhoods together? e.g. perhaps using the DBSCAN-based clusters referenced at bottom of page 16/Supp Fig 3, in lieu of neighborhoods?

Authors' response to reviewers

Reviewer #1: I appreciate the authors' thorough and thoughtful consideration of reviewers' concerns. I have a few remaining questions or notes:

Response: We thank the reviewer for the previous comments and the additional comments.

(1) Fig 1a, is the Y axis scale correct? Looks like DNA DIS scores ceiling out around a value of 4?

Response: We thank the reviewer for the comment. We have changed the Y-axis scale such that the upper limit is 5.

(2) Bottom of page 8, thank you for adding the number of families alongside the number of variants being tested for each proband-sibling combination. Can the authors adopt this consistently throughout the text, e.g. mid-page 9, newly added text noting results from female pro-female sib and female pro-male sib pairs? (and there may be others)

Response: We thank the reviewer for the suggestion. We have added the number of families:

We repeated the analysis for de novo variants from a family with a female proband and a female sibling (n=4,664 from 123 families) and for variants from a family with a female proband and a male sibling (n=3,948 from 101 families)

(3) Supp Fig 7a - X axis labels are gene names, can the authors clarify in the legend that this is how neighborhoods are named, e.g "28 significant neighborhoods labeled by their central gene"

Response: We thank the reviewer for the suggestion. We have added the following description: "Each neighborhood is labeled by its central gene".

(4) Thank you for adding the note about the number of variants shared between neighborhoods. I assume then that there is also considerable sharing of genes between neighborhoods? I apologize for not considering this in depth on the first submission, but what is the rationale for not merging these highly overlapping neighborhoods together? e.g. perhaps using the DBSCAN-based clusters referenced at bottom of page 16/Supp Fig 3, in lieu of neighborhoods?

Response: We thank the reviewer for the question. Using neighborhoods allowed us to more systematically identify subsets of variants potentially associated with the phenotype. We had already established there was an association signal with gene expression sets defined based on genes with tissue differential expression within the brain and were interested in seeing if we could better isolate the potential source of the signal and determine how specific it was. We expected a partition-based clustering would not provide a sufficient level of fine-grained and comprehensive information as the neighborhoods towards these goals. In addition, the

neighborhood sizes are constant, which would not be the case for clustering, therefore avoiding confounding in comparing associations driven by different power from different numbers of variants. However, combining aspects of our framework with clustering based

frameworks such as DBSCAN could be a potentially interesting direction for future work.

We expanded our discussion section to discuss this point:

We also note that ENSAS neighborhood strategy, while allowing systematic coverage of the expression space and removing the confounder of different sizes for variant sets, does lead to a large number of overlapping neighborhoods. While ENSAS supports both Bonferroni- and permutation-based strategies for controlling for multiple testing, future work could investigate alternative approaches to reduce the number and overlap of neighborhoods tested with ENSAS such as through clustering or selecting representative neighborhoods prior to testing.